# Analysis of the transcriptomic and metabolomic landscape of prostate cancer with different anatomical origins using snFLARE-seq and mxFRIZNGRND

Prostate cancer cells of different anatomical locations display remarkable heterogeneity. This poses a challenge to the clinical relevance of pre-clinical models and the efficacy of contemporary therapeutic approaches. Here we develop the snFLARE-seq and mxFRIZNGRND methodologies to directly investigate the transcriptomic and metabolomic landscape of prostate cancer patients utilizing formalin-fixed paraffin-embedded (FFPE) specimens. A retrospective analysis reveals the clinical disparities of prostate cancer from peripheral zone (PZ), transition zone (TZ), and across PZ and TZ. The snFLARE-seq, refined for enhanced single-nucleus sequencing, unveils distinct cell type distributions and signaling pathways between PZ and TZ samples. Hormone therapy substantially affects cancer cells and microenvironment, leading to a polarized feature of epithelial cells and a subverted immune microenvironment. With improvements in metabolite extraction, mxFRIZNGRND reveals unique metabolic features of prostate cancer from different origins. The metabolomic results indicate that PZ cancer cells are in a metabolic-dormant status, which are probably awaken by hormone therapy. Integrative analysis of results from snFLARE-seq, mxFRIZNGRND, and TCGA database uncovers four metabolic pathways and related genes associated with disease aggressiveness. Our work could accelerate investigations on disease heterogeneity and evolution in real-world clinical settings, stimulating patient-specific precision healthcare solutions.

Prostate cancer, the most prevalent malignancy in men, is a global challenge with more than 1.4 million new cases annually[1]. Targeting androgen-androgen receptor (AR) axis is the mainstay for prostate cancer therapy[2–4]. However, due to the intratumoral heterogeneity, treatment resistance is inevitable for some patients[5,6]. Delineating the heterogeneity and evolutionary trajectories of prostate cancer would provide insight to improve patient stratification and develop efficient treatment strategies[7,8].

Around 70% of prostate cancer cases originate from the peripheral zone (PZ) of the prostate gland, with 25% arising in the transition zone (TZ)[9]. As the disease progression, some patients develop extensive lesions across PZ and TZ. Cancer cells from different zones may have unique molecular characteristics and display varying sensitivities to clinical treatment[10–12]. It has been reported that patients with TZ prostate cancer are characterized by higher baseline prostate-specific antigen (PSA) levels and larger cancer volume, but show more favorable clinical outcomes than those with PZ cancer[13]. Despite these intriguing clinical observations, the molecular profiles underlying these anatomical disparities, which can hardly be mimicked by pre-clinical models, remain elusive. As the burden of DNA mutations is not

✉e-mail: hssfline@tongji.edu.cn; dr.ghq@nju.edu.cn; lnchen@sjtu.edu.cn; huiru_tang@fudan.edu.cn; xuefeng_qiu@nju.edu.cn; zhenfei.li@sibcb.ac.cn

substantial in primary prostate cancer, disparities in transcriptome and metabolome are likely to contribute to the multifocal and histological heterogeneity[14].

Formalin-fixed paraffin-embedded (FFPE) blocks, derived from patients with diverse genetic backgrounds and various clinical treatments, provide excellent materials for retrospective research to decipher the intrinsic clonal heterogeneities and identify essential determinants driving malignant evolution[15]. However, the application of FFPE samples in scientific research is hindered by technological challenges. Several approaches based on single-nucleus RNA-sequencing (snRNA-seq) have been developed to analyze FFPE tissues. However, snPATHO relies on probe-based 10X Genomics technology and detects only pre-designed genes[16]; snFFPE, using poly(A)-based 10X Genomics technology, has a low sensitivity in gene detection[17]; and snRandom-seq, derived from the MATQ-Drop technology, may have incorrect sequence alignment and artificially inflated gene counts due to the excessively short fragments for detection[18]. With continuous improvement and optimization of snRNA-seq for FFPE samples, a sophisticated and cost-effective platform could accelerate the analysis of real-world clinical specimens, circumventing the limitations inherent in animal models[19].

The extraction of metabolites from FFPE samples also remains challenging due to the conventional methods used for paraffin removal. Techniques such as thermal deparaffinization, which involves heating FFPE samples to temperatures ranging from 70 to 90 °C, lead to the degradation and loss of heat-sensitive metabolites, potentially skewing metabolic profiles and compromising research outcomes[20-23]. Other approaches based on xylene for paraffin dissolution are associated with the substantial loss of lipid metabolites[24,25]. Advances in

techniques for metabolomics research based on FFPE samples are required to reveal the clinical significance of tumor metabolic features.

In this work, we develop snFLARE-seq (snRNA-sequencing for FFPE samples with Ligation in droplet After in situ REverse transcription) to analyze the transcriptome of different foci within whole-mount prostate specimens. We also develop mxFRIZNGRND (Metabolites extract for FFPE samples using freezing and grinding) for metabolome analysis using FFPE samples. The integration of snFLARE-seq and mxFRIZNGRND could facilitate the discovery of pivotal biological processes and key determinants driving the aggressive progression of prostate cancer from different origins.

## Results

### Clinical responses of prostate cancer from different origins

To investigate the clinical responses of prostate cancer from different origins to prostatectomy, a cohort of 235 patients with localized prostate cancer who received radical prostatectomy between January 2021 and December 2022 was retrospectively analyzed. Pathological examination of whole-mount prostate slices revealed that the index lesion, defined as the lesion with the highest International Society of Urological Pathology (ISUP) grade, was located in PZ for 142 patients (Zone 1 in Fig. 1a), in TZ for 59 (Zone 2 in Fig. 1a), and spanned both PZ and TZ for 34 patients (Zone 3 in Fig. 1a). Compared to patients with PZ lesions, patients with TZ lesions showed a significantly longer biochemical recurrence (BCR)-free survival (hazard ratio [HR] = 0.22, 95% confidence interval [CI] 0.052−0.95, P = 0.042). Notably, patients with lesions across both zones showed significantly shorter BCR-free survival (HR = 2.21, 95% CI 1.009−4.82, P = 0.047) (Fig. 1a, and Supplementary Table 1). These findings indicate that cancers originating from

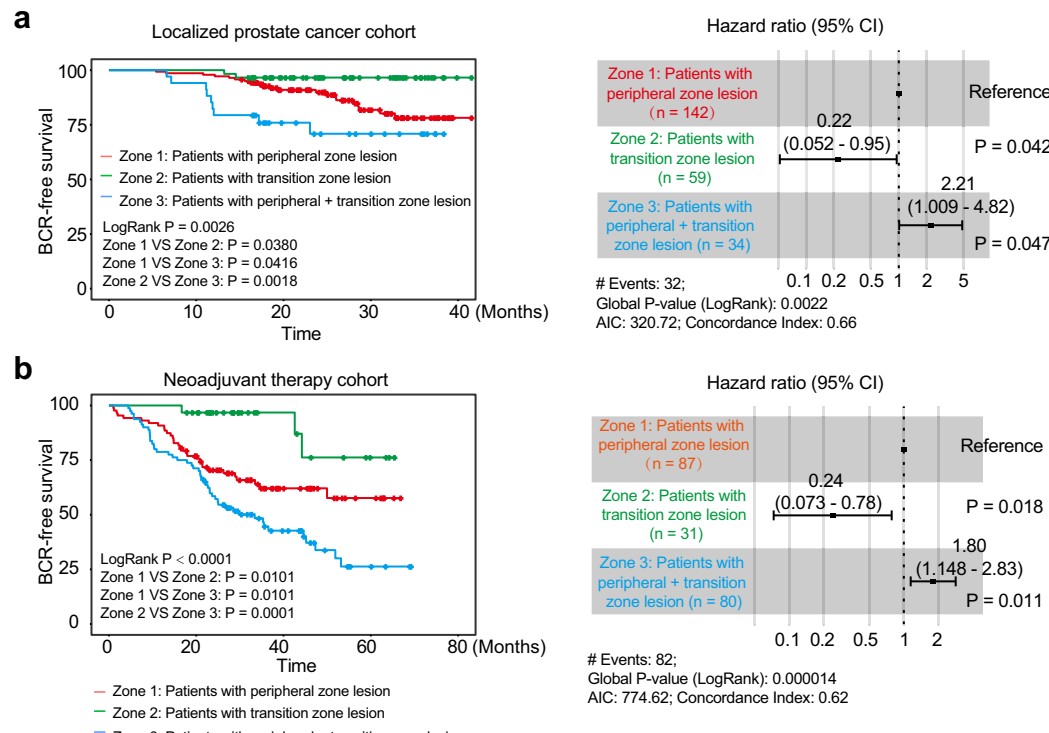

**Fig. 1 | Clinical and pathological sampling overview. a** Correlation of tumor origin with biochemical recurrence (BCR) in localized prostate cancer patients without neoadjuvant therapy. Patients with peripheral zone lesion, *n* = 142; patients with transition zone lesion, *n* = 59; patients with peripheral + transition zone lesion, *n* = 34. **b** Correlation of tumor origin with BCR in localized prostate cancer patients receiving neoadjuvant therapy. Log-rank analysis and multivariate analysis with cox

proportional hazards regression. Patients with peripheral zone lesion, *n* = 87; patients with transition zone lesion, *n* = 31; patients with peripheral + transition zone lesion, *n* = 80. Error bars for hazard ratio indicate hazard ratio and 95% confidence interval (CI). AIC Akaike information criterion. Source data are provided as a Source Data file.

different zones exhibit distinct clinical characteristics in East Asian patients.

To investigate the clinical response to neoadjuvant androgen deprivation therapy (ADT), a cohort of 198 patients who received neoadjuvant ADT prior to radical prostatectomy from September 2017 to October 2022 were assessed. According to the final pathology after radical prostatectomy, 87 patients showed residual lesions in PZ (Zone 1 in Fig. 1b), while 31 in TZ (Zone 2 in Fig. 1b), and 80 with tumors spanning both PZ and TZ (Zone 3 in Fig. 1b). Consistently, patients with TZ lesions had a significantly longer BCR-free survival (HR = 0.24, 95% CI 0.073−0.78, $P$ = 0.018), while those with lesions in both zones had a significantly shorter BCR-free survival (HR = 1.80, 95% CI 1.148−2.83, $P$ = 0.011) (Fig. 1b, and Supplementary Table 2). The results from our single-center retrospective cohort indicate that cancers spanning PZ and TZ might be more aggressive in resisting neoadjuvant ADT.

Parts of FFPE samples were selected for transcriptomic and metabolomic analysis to delineate zone-related tumor heterogeneity and its correlation with disease aggressiveness. PZ lesions (PZ samples, $n$ = 4) and TZ lesions (TZ samples, $n$ = 6), along with their adjacent non-cancerous tissues, were carefully selected from FFPE blocks of 5 patients to elucidate the differences between these zones (Supplementary Fig. 1a). Since most lesions confined only to PZ or TZ diminished remarkably after 6-month of neoadjuvant ADT, four residual malignant lesions spanning PZ and TZ after neoadjuvant ADT (PTM samples, $n$ = 4) were selected as potential "positive controls" for aggressive prostate cancer (Supplementary Fig. 1b).

### Development and validation of snFLARE-seq

To investigate the transcriptomic landscape of prostate cancer from different origins using FFPE samples, snFLARE-seq, based on the single-nucleus sequencing technique was developed with the following optimizations (Fig. 2a): 1. Nuclei isolated from FFPE slices were incubated at 20 °C for 30 min with both proteinase K and SDS, followed by partial crosslink reversal at 80 °C for 15 min. This optimization enhanced the balance between preserving nuclear integrity and maximizing gene detection, as evidenced by an increase of median gene count per cell from 48 to 914 (Supplementary Fig. 2a). 2. In situ reverse transcription within the nuclei was performed using a mixture of random primers and oligo-dT primers, resulting in an over fivefold increase in the number of detectable genes (Supplementary Fig. 2b; 89 vs 489 median gene/cell). 3. After in situ reverse transcription, cDNA was tagged with a unique cell barcode within droplets. A secondary template switching reaction was conducted using reverse transcriptase after breaking the oil and thorough crosslink reversal. Completing thorough crosslink reversal before proceeding with the template switching reaction effectively minimized the interference of RNA crosslinking with the template switching process (Supplementary Fig. 2c; 652 vs 777 median gene/cell). Finally, index PCR was employed to construct a cDNA library for further sequencing.

A standard mixed-species validation experiment for snFLARE-seq was conducted using FFPE human renal carcinoma samples and FFPE mouse renal samples. A low doublet rate (less than 2.31%) was observed, with only 183 nuclei, among 7932 detected, aligned to both human and mouse genomes (Supplementary Fig. 3a). To further assess the performance of snFLARE-seq, we compared it to snRNA-seq using two fresh mouse kidney samples and two FFPE mouse kidney samples. In fresh kidney samples with regular 3′ snRNA-seq, a median of 817 and 818 genes per nucleus were detected, with sequencing saturation at 41.62% and 43.10%, respectively. snFLARE-seq yielded a median of 786 and 578 genes per nucleus from FFPE samples, with sequencing saturation at 38.34% and 38.25%, respectively (Supplementary Fig. 3b). The unique molecular indices (UMIs) counts were also found to be comparable between the two methods (Supplementary Fig. 3b). A total of 16,119 cells were clustered into 15 distinct cell types, according to the results of snFLARE-seq (Supplementary Fig. 3c, d). The cell types

identified in FFPE kidney samples using snFLARE-seq were largely consistent with those determined by conventional snRNA-seq (Supplementary Fig. 3e, f). Additionally, the gene expression profiles obtained from the two methods were found to be relatively consistent (Supplementary Fig. 3g).

We also compare snFLARE-seq results with the reported data of other FFPE-based approaches. Compared with probe-based scFFPE-seq, snFLARE-seq exhibited superior gene detection capacity (Supplementary Fig. 4a). In terms of cellular composition, epithelial cells dominated the clusters identified by scFFPE-seq (sample ID: Lung_Cancer_Manual_BC), whereas snFLARE-seq revealed a broader spectrum of immune cells and fibroblasts (Supplementary Fig. 4b). For snRandom-seq, FFPE samples were only treated with proteinase K without pre-crosslinking reversal, thus the reads length of RNA fragments from snRandom-seq is shorter, while more than 90% of fragments from snFLARE-seq were longer than 50 bp (Supplementary Fig. 4c, d). To further compare our method with snRandom-seq, the same pre-crosslink reversal scheme was utilized to obtain nucleus samples followed with two different single-cell library preparation protocols by using FFPE human liver cancer samples. A total of 489 genes and 685 UMIs were detected with sequencing saturation at 28.25% by snFLARE-seq; whereas 314 genes and 366 UMIs were detected with sequencing saturation at 35% by snRandom-seq (Supplementary Fig. 4e). Random primers, together with oligo-dT, were used for snFLARE, snRandom, and scFAST-seq, but not snFFPE-seq. Thus, reads from snFFPE-seq were more enriched in the 3′ end while reads from snFLARE-seq covered the whole gene body, making it feasible to detect non-coding RNAs, RNA splicing, or gene mutations (Supplementary Fig. 4f, g).

We extended the validation of snFLARE-seq to assess its capability across a range of human tissues. A random selection of FFPE samples from human endometrial cancer, colorectal cancer, kidney cancer, liver cancer, and lung cancer was subjected to testing. All tissue type samples achieved median genes per cell count of above 900, indicating the robustness of snFLARE-seq across various malignancies (Supplementary Fig. 4h). By analyzing all 11 samples we have tested for snFLARE-seq, the sensitivity metric of median genes per cell was found to be closely correlated with the RNA content and size distribution within the FFPE samples (Supplementary Fig. 4i). Consequently, we established the following criteria for FFPE samples suitable for snFLARE-seq: total RNA extracted from 20,000 nuclei should be ≥25 ng, while the main peak of RNA fragments is ≥200 nucleotides.

### snFLARE-seq results for prostatic specimens

A total of 101,729 cells were successfully sequenced, using 14 FFPE samples from 9 patients (Supplementary Fig. 1). Cells were subjected to unsupervised clustering into 13 distinct cell types using uniform manifold approximation and projection (UMAP) analysis (Fig. 2b). All clusters were further annotated based on typical cell type markers (Supplementary Fig. 5)[26–29]. Overall, PTM samples exhibited greater divergence from PZ and TZ samples, indicating that clinical interventions substantially remodeled cell fate within prostate tissue (Fig. 2c). Odds ratios (OR) analysis showed that fibroblasts and epithelial cells were predominantly enriched in TZ samples, whereas PZ samples were predominantly composed of epithelial cells, consistent with previous report[30,31]; an enrichment of nerve cells, pericytes, and proliferating cells, alongside a notable reduction in epithelial cells was observed in PTM samples (Fig. 2d)[32]. The distribution of cell numbers and clusters for each patient is shown in Fig. 2e. These findings highlight the heterogeneity of cell subpopulations within these three cohorts.

### Polarized status of epithelial cells after clinical interventions

Unsupervised clustering of epithelial cells yielded 12 different clusters, which were further annotated using classic cell type markers (Fig. 3a, and Supplementary Fig. 6). To further characterize the transcriptomic

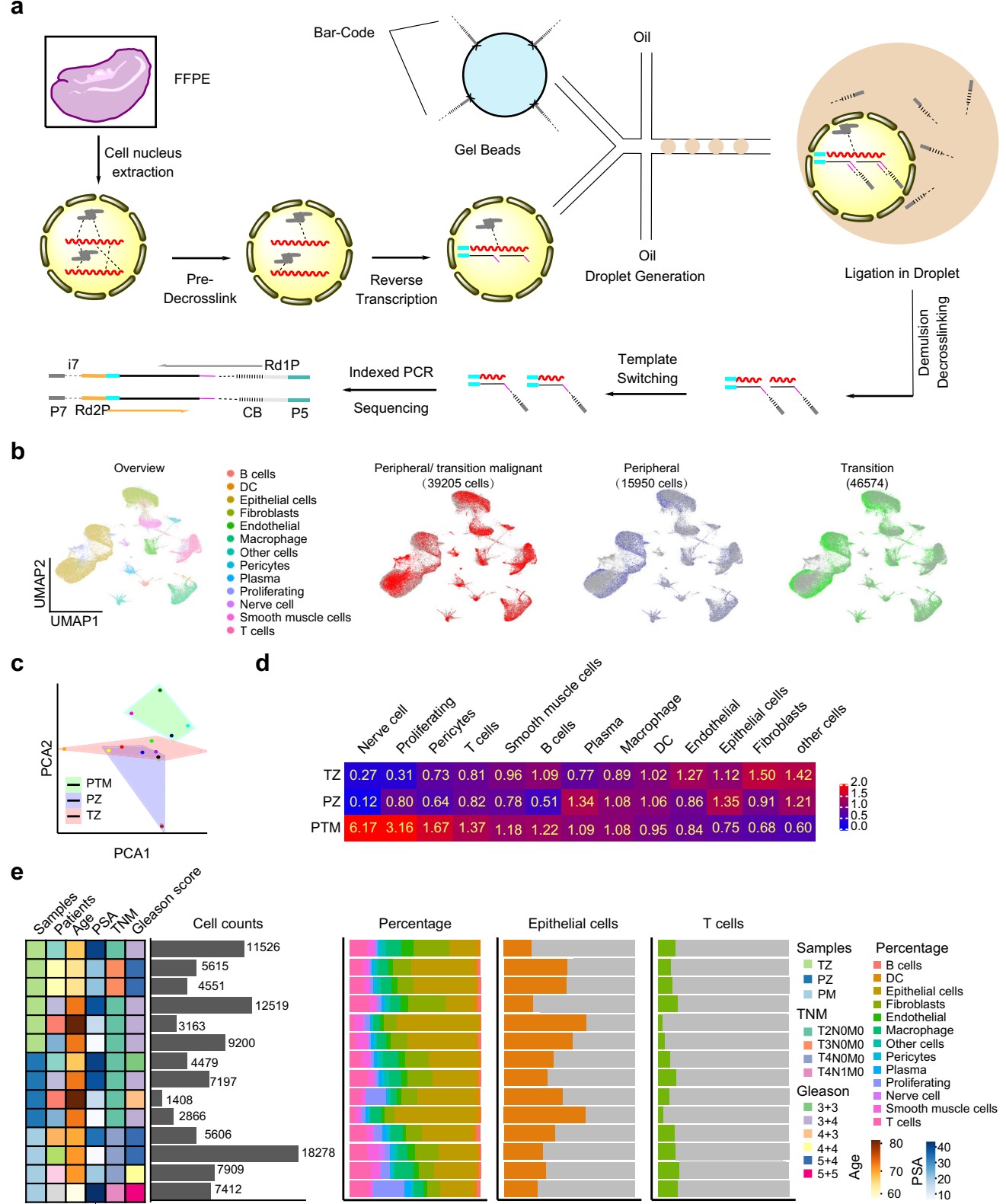

**Fig. 2 | Overall results of snFLARE-seq for prostate cancer of different origins.** **a** Schema for snFLARE-seq. Figures were generated using ChemDraw 18.0. **b** UMAP results of snFLARE-seq from all patients (101,729 cells from 14 samples). Canonical correlation analysis in Seurat package for removal of batch effect and standard Seurat workflow with default parameters for clustering. **c** Principal component analysis (PCA) of samples from different patient cohort. PZ, patients with tumor origin of peripheral zone ($n = 4$); TZ, patients with tumor origin of transition zone ($n = 6$); PTM, patients receiving neoadjuvant therapy with tumor origin across peripheral and transition zone ($n = 4$). **d** Enrichment of different cell types in three cohorts. Odds ratio (OR) values were calculated to assess enrichment of each cluster in specific tissues; an OR > 1.5 indicated a preferential association of specific cell cluster with tissue, whereas an OR < 0.5 indicated a marked depletion. **e** Cell composition distribution for each sample. Each row represents an individual patient. DC dendritic cells.

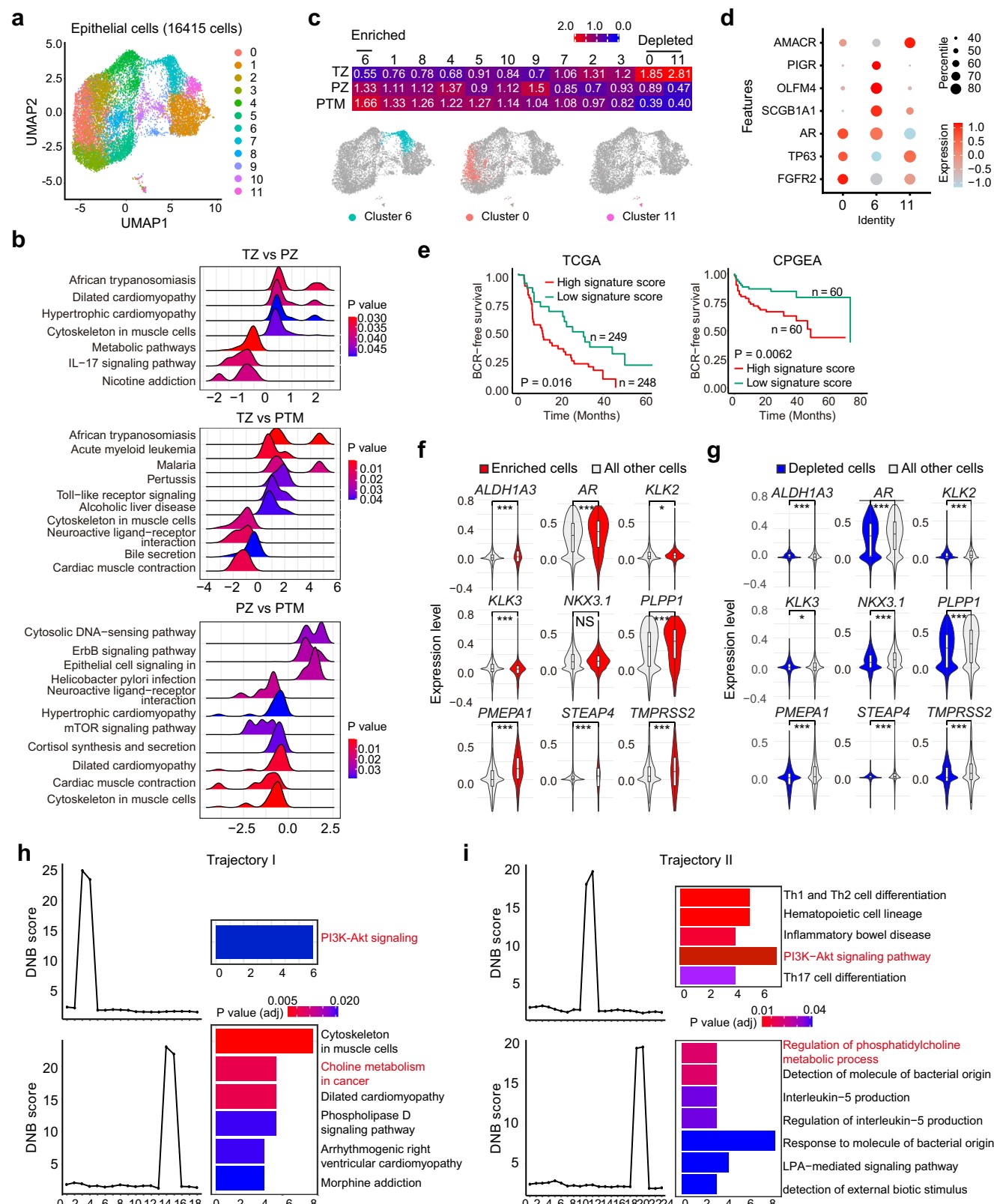

profiles of these cohorts, gene set enrichment analysis (GSEA) was conducted based on differentially expressed genes (Supplementary Fig. 7). IL-17 signaling and metabolic pathways were more activated in PZ samples, compared to TZ samples (Fig. 3b)[33,34]. PTM samples exhibited increased heterogeneity, with pathways such as dilated cardiomyopathy, cytoskeleton in muscle cells, and hypertrophic cardiomyopathy pathways being upregulated compared to PZ samples,

indicating a TZ-like transcriptomic feature. Consistently, cytoskeleton in muscle cells pathway was even more activated in PTM samples than in TZ samples. On the other hand, PTM samples obtained a suppressed African trypanosomiasis pathway, indicating a PZ-like transcriptomic feature. Notably, a suppressed cytosolic DNA-sensing pathway and an activated cortisol synthesis and secretion pathway were found in PTM samples, suggesting an inactive innate immune system and enhanced

**Fig. 3 | Characteristics of epithelial cells within three cohorts. a** UMAP results of epithelial cells from all patients (16,415 cells from 14 samples). **b** Gene set enrichment analysis (GSEA) of epithelial cells from different cohorts. PZ, 3895 cells from 4 samples; TZ, 9049 cells from 6 samples; PTM, 3471 cells from 4 samples. The statistical significance of enrichment was evaluated using GSEA, with a nominal $P < 0.05$ and a false discovery rate (FDR) $< 0.25$ considered statistically significant. **c** Enrichment of different clusters in three cohorts. PZ, 3895 cells from 4 samples; TZ, 9049 cells from 6 samples; PTM, 3471 cells from 4 samples. The positions of the enriched (OR > 1.5) or depleted (OR < 0.5) clusters were shown in the UMAP plot through in situ staining. **d** The characteristic markers for cluster 0, 6, and 11. Cluster 0, 2808 cells from 14 samples; Cluster 6, 1126 cells from 14 samples; Cluster 11, 102 cells from 14 samples. **e** Cluster 6 signature for patient stratification in TCGA and

CPGEA databases. High signature score patients in TCGA, 248; low signature score patients in TCGA, 249; High signature score patients in CPGEA, 60; low signature score patients in CPGEA, 60. Log-rank test for Kaplan–Meier survival analysis. **f, g** Expression of androgen response genes in the enriched or depleted clusters within PTM samples (3471 cells from 4 samples). Violin plots depict the kernel density of single-cell expression values. Median, 25th percentile, and 75th percentile are shown in the box plot; whiskers extending to 1.5 X IQR (interquartile range). **h, i** Dynamic network biomarkers (DNB) analysis for the most drastic changes across gene expression network of epithelial cells in trajectory I and II (16,415 cells from 14 samples). Two-side pairwise Wilcoxon signed-rank test with Benjamini–Hochberg for $P$ value correction. *, ** and *** denoted $P < 0.05$, $P < 0.01$ and $P < 0.001$, respectively.

glucocorticoid receptor (GR) function in PTM samples, which may confer treatment resistance (Fig. 3b).

OR analysis revealed that cluster 9 (also annotated as Ki67⁺ Club cell-2 in Supplementary Fig. 6b) was more enriched in PZ samples, while cluster 0 (FGFR2 high expression Basal cell-1 in Supplementary Fig. 6b) and cluster 11 (AMACR⁺ AR⁻ CRPC-2 in Supplementary Fig. 6b) were more prevalent in TZ samples (Fig. 3c)[35,36]. The cell type distribution in PTM samples was more closely aligned with PZ samples than with TZ samples, exhibiting a polarized status of PZ epithelial cells after clinical interventions: cell subtypes which are less prevalent in PZ samples get even scarcer in PTM samples, while the abundant cell subtypes in PZ samples are further enriched in PTM samples. Specifically, cluster 6 (SCGB1A1⁺ OLFM4⁺ Club cell-1 in Supplementary Fig. 6b) was enriched, while cluster 0 and 11 were depleted in PTM samples (Fig. 3c, d)[37,38]. Consistently, club cells were also linked to prostate cancer aggressiveness previously (Supplementary Fig. 6c, d)[39,40]. A signature derived from the transcriptome of cluster 6 successfully distinguished aggressive prostate cancer patients in TCGA and Chinese Prostate Cancer Genome and Epigenome Atlas (CPGEA) databases (Fig. 3e)[41]. Thus, these epithelial clusters may explain why PZ prostate cancer is more aggressive than TZ prostate cancer. Comparing to all the other clusters, the enriched cluster (cluster 6) exhibited enhanced IL-17 signaling pathways, whereas the depleted clusters (0 and 11) showed suppressed IL-17, TLR, TNF, and NF-κB signalings, providing hints for PZ prostate cancer specific therapeutic strategies (Supplementary Fig. 8). The AR pathway signature-related genes were further examined in the enriched cluster and the depleted clusters[42]. AR and its downstream effectors were upregulated in the enriched cluster, but suppressed in the depleted clusters, indicating that neoadjuvant therapy may not effectively eliminate androgen-responsive cells but rather selects for cells with potent AR pathway activity (Fig. 3f, g). Collectively, these results together suggest that prostate cancer of different origins are heterogeneous at the transcriptome level, and the differentiated signaling pathways provide potential targets for zone specific prostate cancer therapy and circumventing resistance to hormone therapy.

### Potential intervention approaches targeting epithelial cells

To find potential intervention strategies for PZ, especially PTM samples, drug response analysis was conducted based on transcriptomic features of epithelial cells and the enriched cluster within PTM samples. Hyperforin, a multifunctional natural compound with potential functions to regulate IL-17 pathways, Ca²⁺ channels, AKT signaling, and other activities, emerged as a potential therapeutic agent to treat PTM (Supplementary Fig. 9a, b)[43–46]. IL-17 pathway has been reported to regulate PD-L1 expression in prostate cancer or create an immunotolerant tumor microenvironment to facilitate the progression of prostate cancer at different disease stages[47–50]. Consistently, validation experiments have been conducted to reveal that hyperforin suppressed cell growth in LNCaP, C4-2, and VCaP cells (Supplementary Fig. 9c).

The evolutionary trajectory of cluster 6 was also investigated to find potential intervention strategies. Given the similarities between PTM epithelia cells and PZ epithelial cells, we integrated epithelial cells from PZ and PTM samples for evolution analysis. Cluster 9, which is enriched in PZ samples, was used as the starting point to construct a pseudotime trajectory. Two distinct evolutionary trajectories leading to cluster 6, which is enriched in PTM samples, were established (Supplementary Fig. 10a). Sequentially expressed genes associated with these two trajectories were identified (Supplementary Fig. 10b). Trajectory 1 was divided into 18 time windows using a sliding window approach, with each window consisting of 100 cells and a step size of 50 cells. Dynamic network biomarkers (DNB) analysis was used to detect the most drastic changes of gene expression networks in cells, thereby pinpointing fate-decision points[51–58]. Window 3 (transition from cluster 9 to cluster 4) and window 14 (transition from cluster 4 to cluster 6) were identified as critical fate-decision points by DNB analysis. Genes related to PI3K-AKT signaling substantially changed in window 3, followed by marked alterations in pathways such as choline metabolism in cancer in window 14 (Fig. 3h). Consistently, the sequential modulation of PI3K-AKT signaling and the regulation of phosphatidylcholine metabolism process were also observed for trajectory II (Fig. 3i). PI3K-AKT pathways have been identified to promote the development of prostate cancer and provide resistance to hormone therapy[59–61]. Choline metabolism pathway has also been reported as a potential therapeutic target for prostate cancer[62,63]. Our data here indicate that the perturbation of PI3K-AKT signaling and choline-related metabolism are crucial for disease progression and aggressive evolution, and might be targeted for early intervention.

### Subverted immune microenvironment in PTM samples

Immune cells, including T cells, macrophages, and dendritic cells (DCs), across three cohorts were further analyzed. T cells were clustered into 10 distinct subpopulations (Fig. 4a, and Supplementary Fig. 11a, b). PZ and TZ samples shared a similar distribution of T cell subsets, with an enrichment of cluster 5 (RORA⁺ tissue-resident memory T cells, in Supplementary Fig. 11b) and cluster 6 (CD4⁺ T central memory, TCM, in Supplementary Fig. 11b). Notably, TZ samples had a higher proportion of cluster 8 (CD8⁺ effector cells with high ITGA1 expression, in Supplementary Fig. 11b) and cluster 9 (CD4⁺ Th17 cells, in Supplementary Fig. 11b), compared to PZ samples (Fig. 4a). However, the immune environment dramatically transformed in PTM samples. The enriched clusters (5, 6, 8 and 9) in PZ/TZ samples experienced a notable depletion in PTM samples, whereas cluster 2 (CD4⁺ Th) and 4 (CD4⁺ Treg) became predominant (Fig. 4a). GSEA analysis also showed a comparable transcriptional feature between PZ and TZ, indicating that T cells could migrate across different zones to create a relatively homogeneous immune environment (Supplementary Fig. 12a). However, PTM samples displayed pronounced transcriptional differences, with plenty of pathways altered (Supplementary Fig. 12b, c). Specifically, transcriptomic features in the enriched or depleted T cell clusters of PTM samples were analyzed.

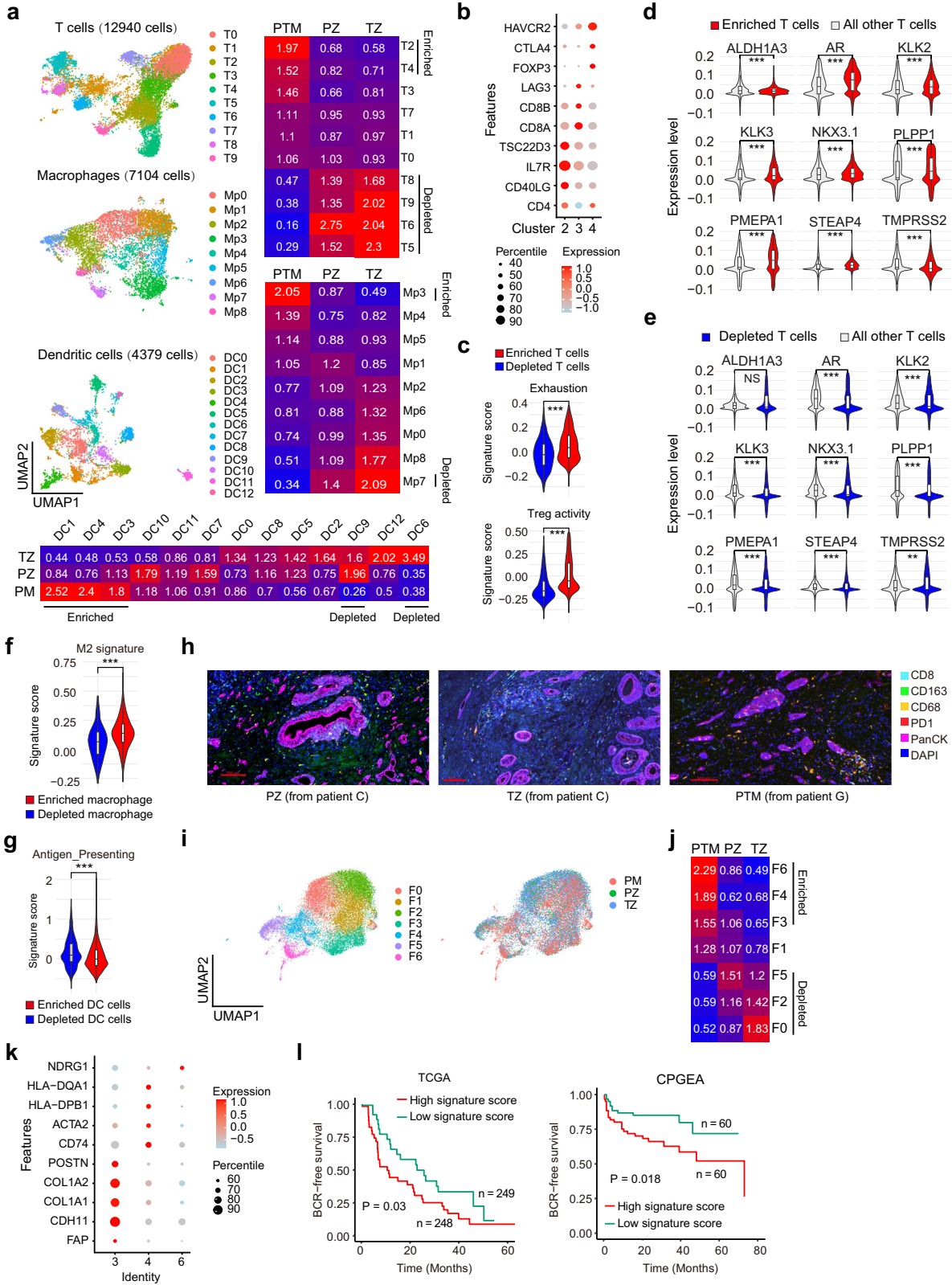

GSEA results revealed that pathways, including Th17 cell differentiation and measles pathways, were activated in the enriched clusters (cluster 2 and 4), but suppressed in the depleted clusters (cluster 5, 6, 8 and 9). In contrast, pathways including ECM-receptor interaction, focal adhesion, and PI3K-AKT signaling, were suppressed in the enriched clusters but activated in the depleted clusters (Supplementary Fig. 12d). The characteristic markers for clusters 2, 3, and 4 were

illustrated in Fig. 4b. Enriched T cells exhibited activation of pathways associated with T cell exhaustion and regulatory T cell (Treg) activity, collectively suggesting the presence of a suppressive immunological microenvironment (Fig. 4c)[39]. Consistently, exhausted T cells and Treg cells have been previously linked to prostate cancer aggressiveness (Supplementary Fig. 13)[39,40]. Androgen and AR have been well-documented for their immunosuppressive effects[64,65]. Our analysis

**Fig. 4 | Characteristics of microenvironment in three cohorts. a** UMAP results and enrichment of immune cells from all patients. Clusters of T cells (12,940 cells from 14 samples), macrophage (7104 cells from 14 samples), and dendritic cells (4379 cells from 14 samples) were analyzed. Cluster distributions in different cohorts were displayed. **b** The characteristic markers for cluster 2, 3, and 4. Cluster 2, 1869 cells from 14 samples; Cluster 3, 1707 cells from 14 samples; Cluster 4, 1370 cells from 14 samples. **c** Pathways involving T cell exhaustion and regulatory T cell (Treg) activity in enriched and depleted T cells. Enriched cells, 3239 cells from 14 samples; Depleted cells, 2597 cells from 14 samples. **d, e** Expression of androgen response genes in the enriched or depleted T cell clusters from PTM samples. Enriched cells, 3239 cells from 14 samples; Depleted cells, 2597 cells from 14 samples. **f** Pathways involving M2 activity in enriched and depleted macrophage clusters. Enriched cells, 1107 cells from 14 samples; Depleted cells, 266 cells from 14 samples. **g** Pathway involving antigen presenting activity in enriched and depleted DC clusters. Enriched cells, 1269 cells from 14 samples; Depleted cells, 565

cells from 14 samples. Violin plots depict the kernel density of single-cell expression values. Median, 25th percentile, and 75th percentile are shown in the box plot; whiskers extending to 1.5 X IQR. **h** Multicolor IF staining shows the distributions of immune cells and epithelial cells in PZ, TZ, and PTM samples. Scale bar, 100 μm. Representative images are shown (*n* = 2 samples per group). **i** UMAP results of fibroblast cells (17,826 cells from 14 samples) from all patients. **j** Cluster distribution of fibroblast cells in different cohorts. PZ, 2620 cells from 4 samples; TZ, 9635 cells from 6 samples; PTM, 5572 cells from 4 samples. **k** The characteristic markers for fibroblast cluster 3, 4, and 6. Cluster 0, 2808 cells from 14 samples; Cluster 4, 1812 cells from 14 samples; Cluster 6, 1087 cells from 14 samples. **l** Signature of enriched fibroblast clusters for patient stratification in TCGA and CPGEA databases. High signature score patients in TCGA, 248; low signature score patients in TCGA, 249; High signature score patients in CPGEA, 60; low signature score patients in CPGEA, 60. Two-side pairwise Wilcoxon signed-rank test with Benjamini−Hochberg for *P* value correction. *, ** and *** denoted *P* < 0.05, *P* < 0.01 and *P* < 0.001, respectively.

revealed that the AR pathway activity was enhanced in the enriched T cell clusters, but suppressed in the depleted clusters for PTM samples (Fig. 4d, e). Together, these data demonstrate that clinical interventions lead to a subverted immune environment in the prostate tissue.

Consistently, macrophages and DCs exhibited relatively homogeneous cluster distributions between PZ and TZ samples, whereas PTM samples displayed a distinct heterogeneity (Fig. 4a, and Supplementary Fig. 14). Enriched macrophage clusters in PTM samples had activated cholesterol metabolism, glycerolipid metabolism, and ferroptosis pathways, indicating ferroptosis related drugs may be used following hormone therapy to counteract the immunosuppressive environment (Supplementary Fig. 15a)[66]; Depleted macrophage clusters in PTM samples had activated T cell receptor signaling pathway, PD-L1 expression and PD-1 checkpoint pathway in cancer, and Th17 cell differentiation pathways (Supplementary Fig. 15b). AR signaling was also activated in the enriched macrophage clusters but suppressed in the depleted clusters (Supplementary Fig. 15c, d). Comparing to the depleted macrophage clusters, the enriched clusters displayed enhanced M2 signature (Fig. 4f)[39]. Consistently, M2 signature has also been associated with disease aggressiveness in other databases (Supplementary Fig. 15e, f)[39,40]. For DCs, cholesterol metabolism was also found to be activated in the enriched clusters, indicating a potential correlation of cholesterol metabolism with suppressed immune activity (Supplementary Fig. 16a)[67,68]. Pathways related to aldosterone, cortisol, and progesterone were suppressed in the depleted DC clusters, indicating a potential correlation of these steroids with immune dysfunction (Supplementary Fig. 16b). AR signaling was also activated in the enriched DC clusters, although its role in the depleted DC clusters is a little bit controversial (Supplementary Fig. 16c, d). Comparing to the depleted DC clusters, the enriched clusters displayed suppressed antigen-presenting activity (Fig. 4g)[39]. Results for multicolor staining indicated that more CD8+ T cells were found in TZ samples but less infiltration occurred, comparing to PZ samples (Fig. 4h, and Supplementary Data 1). For PTM samples, CD8+ T cells decreased while PD1-positive cells and M2 macrophage increased, especially in tumor areas (Fig. 4h, and Supplementary Data 1). Overall, these results together indicate that hormone therapy remodels the immune microenvironment dramatically towards a suppressive status.

### Remodeling of fibroblast cells by clinical interventions

Fibroblast cells across the three different cohorts were also analyzed and divided into 7 different clusters (Fig. 4i, and Supplementary Fig. 17). Similar to the immune cells, PZ and TZ samples displayed relatively homogeneous cluster distributions. However, PTM samples had a markedly contrasting distribution profile (Fig. 4j). Cluster 0, 2, and 5 were predominantly found in PZ/TZ samples, but diminished markedly in PTM samples. Cluster 6, 4, and 3, which were minor in PZ/TZ samples, became dominant in PTM samples (Fig. 4j). The

characteristic markers for cluster 3, 4, and 6 were illustrated in Fig. 4k. A signature derived from the transcriptome of the enriched clusters successfully distinguished aggressive prostate cancer patients in TCGA and CPGEA databases (Fig. 4l). GSEA analysis indicate that Hippo signaling pathway, osteoclast differentiation, B cell receptor signaling pathway, and others were activated in the enriched fibroblast clusters, but suppressed in the depleted clusters (Supplementary Fig. 18a). Consistently, AR signaling pathway was activated in the enriched fibroblast cells but suppressed in the depleted cells, indicating that clinical treatment ultimately selects for cells with robust AR activity (Supplementary Fig. 18b, c). Together, these data demonstrate that the cancer-associated fibroblast microenvironment is significantly remodeled by anti-androgen treatment.

### Dysregulated cell communications in PTM samples

We extended our analysis to examine cell communication among different cell types across three cohorts using CellChat[69]. Overall, cell communication profiles in PZ and TZ samples were similar, with epithelial and fibroblast cells emerging as the main signal sources and also the main signal targets (Fig. 5a, b). Meanwhile, the interaction strength markedly reduced in PTM samples. A reduction in active communication between epithelial and fibroblast cells in PTM samples was observed. T cells were more active in signaling to epithelial cells and macrophages, and in responding to signals from fibroblast cells (Fig. 5c). Cell communication pathways, derived from CellPhoneDB, were analyzed in different cohorts[70]. Cell communications involving SEMA4, CD6, and DHEA pathways were more pronounced in PTM samples, while those involving PTPR, CNTN, ADGRL, and UNC5 pathways were enriched in PZ/TZ samples but not PTM samples (Fig. 5d, e). A signature (26 genes including SEMA3A, SEMA3B, SEMA3C, SEMA3D, SEMA3E, SEMA3F, SEMA4A, SEMA4C, SEMA4D, PLXNA1, PLXNA2, PLXNA3, PLXNA4, PLXNB1, PLXNB2, PLXNB3, PLXND1, PPARA, CD6, CD72, ALCAM, UBASH3B, ESR1, NRP1, NR1I2, and NRP2) derived from four pathways (SEMA4, SEMA3, CD6, and DHEA pathways), effectively identified aggressive prostate cancer patients in TCGA and CPGEA databases (Fig. 5f). Surprisingly, this signature is conserved across various cancer types for patient stratification (Fig. 5g). Collectively, these results demonstrate that dysregulated cell communications correlate with disease aggressiveness, and intervention in these communication pathways may offer novel therapeutic avenues for prostate cancer.

### Development of mxFRIZNGRND for metabolomics research with FFPE samples

To investigate the metabolomics in prostate cancer patients with FFPE samples, we have developed a better extraction method, termed the Freezing extraction method (F-extraction), which involves cryogenic pulverization of FFPE samples using liquid nitrogen, followed by

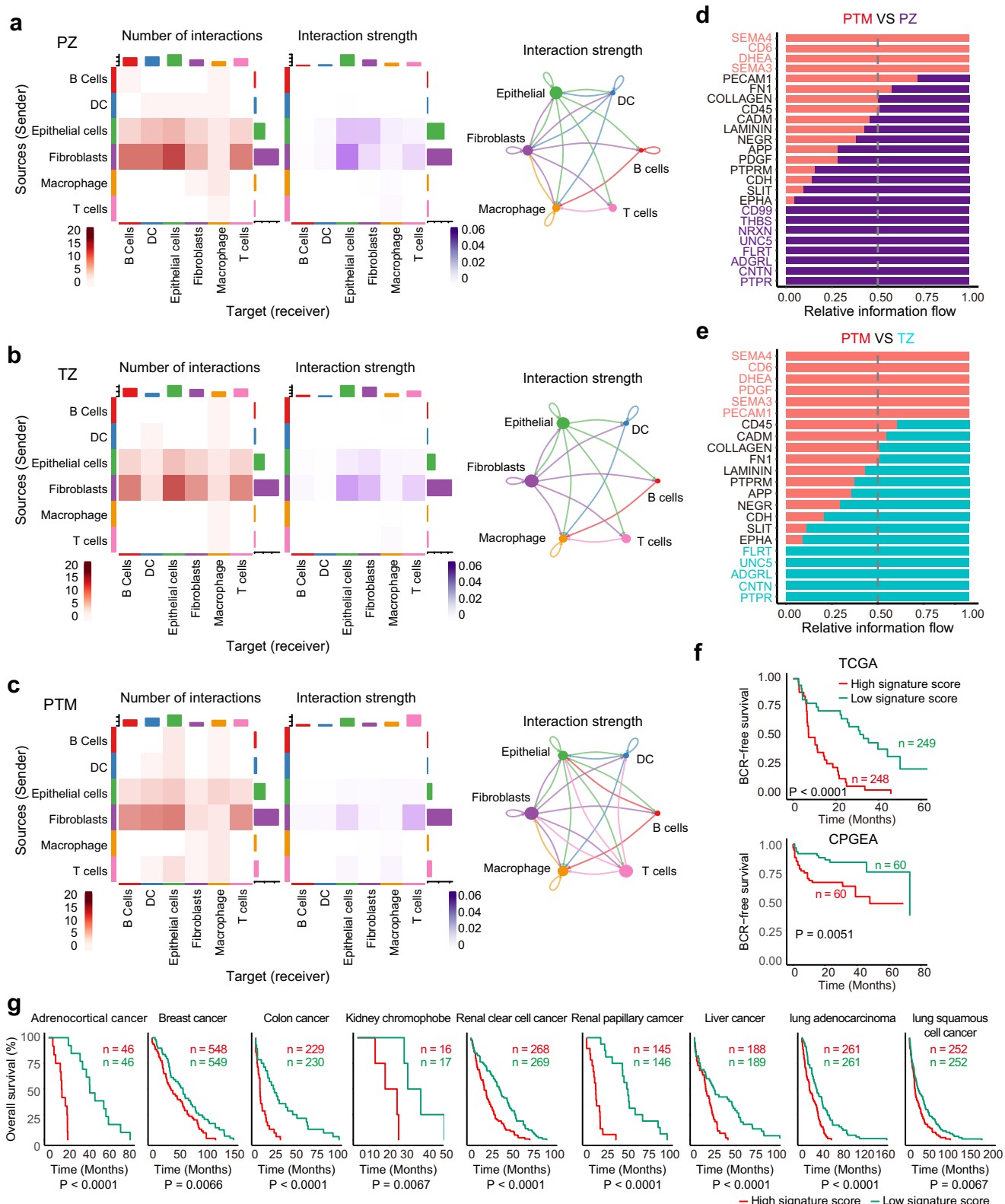

mechanical grinding to facilitate metabolite extraction. Two established extraction methods, the heating for paraffin removal extraction method (H-extraction) and the xylene-based paraffin dissolution extraction method (X-extraction), were also used to extract metabolites from FFPE samples (Fig. 6a)[20,22–25]. FFPE liver samples from mouse were used for this comparative methodological analysis. With most metabolites detected consistently by three methods, the F-extraction demonstrated the highest yield of lipids and hydrophilic metabolites

(Fig. 6b, c). Notably, the MS results revealed that lipids detected by H-extraction exhibited obvious differences from those detected by our F-extraction (Supplementary Fig. 19a). This is likely attributable to the degradation of the heat-sensitive analytes due to heating[21]. The distribution patterns of the detected lipids were further analyzed, and the results showed that less high-molecular weight compounds, but more lower-molecular weight metabolites were detected by H-extraction (Supplementary Fig. 19b). Specifically, a marked reduction in

**Fig. 5 | Cell communications between different cell types in three cohorts.**
Heatmap of interaction number and strength in PZ (**a**; 10,395 cells from 4 samples), TZ (**b**; 29,811 cells from 6 samples), and PTM samples (**c**; 20,094 cells from 4 samples). *Y*-axis, the source cell type of the communication; *X*-axis, the target cell type of the communication. The chord diagram on the right visualizes the communication strength between different cell types, with the thickness of the lines indicating the strength of the communication. **d** Differentially enriched pathways between PTM and PZ samples. Cell communication pathways are derived from CellPhoneDB. **e** Differentially enriched pathways between PTM and TZ samples. **f** Communication signature for patient stratification in TCGA and CPGEA databases.

Communication signature was established based on SEMA4, CD6, and DHEA communication pathways from CellPhoneDB. High signature score patients in TCGA, 248; low signature score patients in TCGA, 249; High signature score patients in CPGEA, 60; low signature score patients in CPGEA, 60. **g** Cell communication signature for patient stratification in different cancer types. Log-rank test for Kaplan–Meier survival analysis. The sequential numbers of high signature score patients in TCGA with indicated cancer types are 46, 548, 229, 16, 268, 145, 188, 261, and 252. The sequential numbers of low signature score patients in TCGA with indicated cancer types are 46, 549, 230, 17, 269, 146, 189, 261, and 252. A significance threshold of *P* < 0.05 was applied.

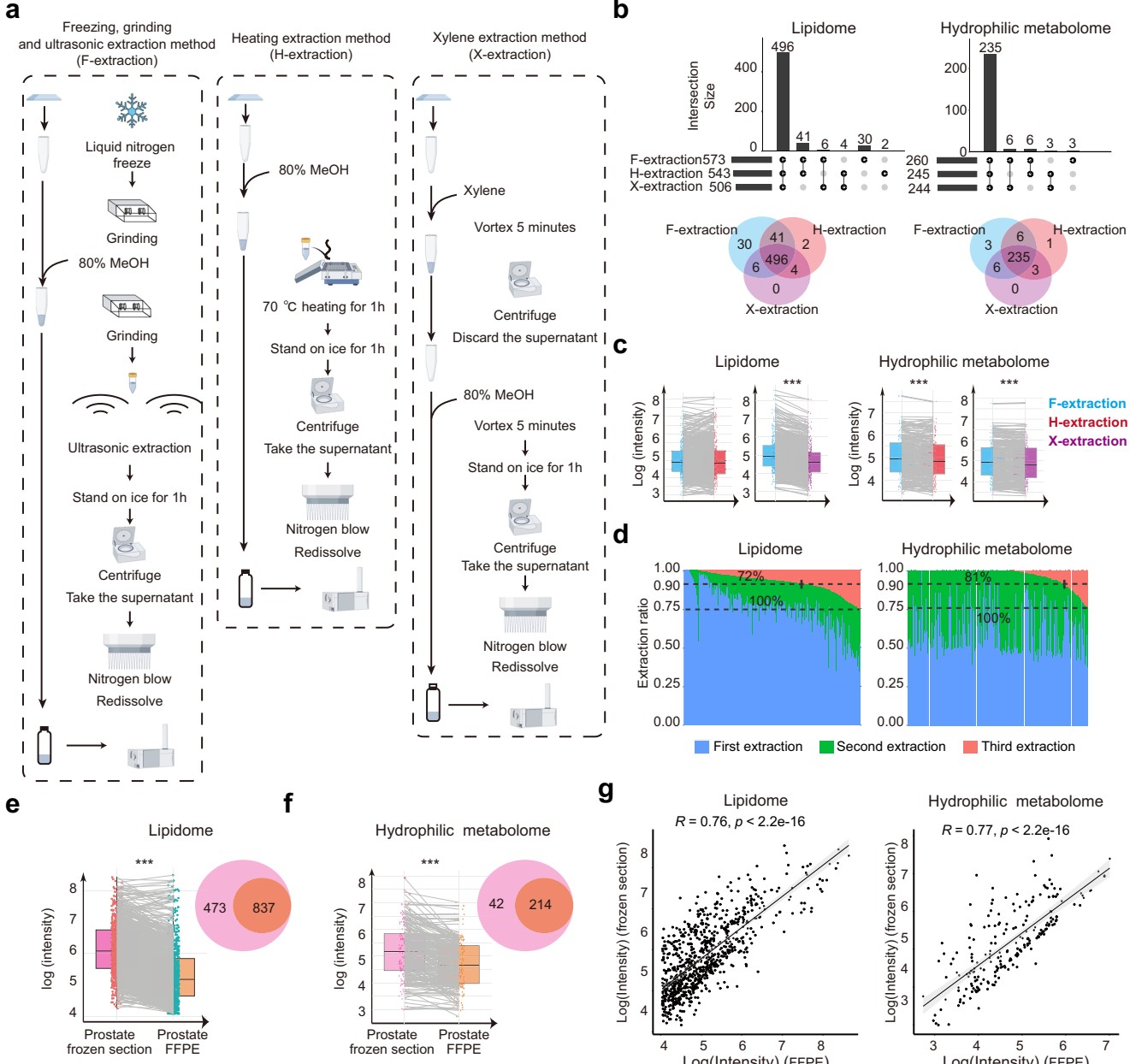

**Fig. 6 | Establishment of mxFRIZNGRND method for metabolome analysis using FFPE samples. a** Flow charts of three extractions for metabolites extraction from FFPE samples. Figures were generated using Figdraw platform (www.figdraw. com). **b** The numbers of detected lipids and hydrophilic metabolites by different extraction methods (*n* = 5 per method). **c** The average analytes' intensity (*n* = 5) obtained from different extraction methods. Two-side pairwise Wilcoxon signed-rank test. **d** Effects of extractions on analytes' recovery. **e**, **f** Comparison of lipids

and metabolites detected from frozen fresh samples (*n* = 3) and FFPE samples (*n* = 3) from mouse prostate tissues. Two-side pairwise Wilcoxon signed-rank test. **g** Correlations of lipids and metabolites detected from frozen fresh (*n* = 3) and FFPE samples (*n* = 3). For box plot, median, 25th percentile, and 75th percentile are shown; the whiskers extend to the 2.5th and 97.5th percentiles. Pearson correlation analysis. *P* < 0.05; **P* < 0.01; ***P* < 0.001.

triglycerides (TG), phosphatidylcholine (PC), and sphingomyelin (SM), along with a marked increase in diacylglycerol (DG), phosphatidylethanolamine (PE), lysophosphatidylcholine (LPC), lysophosphatidylethanolamine (LPE), and sphingosine (Sph), was observed (Supplementary Fig. 19b). These results together support the superiority of our F-extraction for metabolite extraction from FFPE samples.

To optimize the balance between extraction efficiency and the preservation of metabolite integrity, we further refined the F-extraction by incorporating additional experimental parameters, including extraction iterations and aqueous/organic phases separation. Our results indicated that two rounds of methanol extraction efficiently extracted the majority of metabolites from FFPE tissues (more than 90% of analytes with recoveries >70%), ensuring comprehensive metabolite coverage (Fig. 6d). The extract was then mixed with water and methyl tert-butyl ether (MTBE), enabling the sequential isolation of hydrophilic metabolites and lipids from the aqueous and organic phases, respectively. With the improvements in sampling F-extraction, metabolites extraction, and sequential aqueous/organic phases separation, we successfully established the mxFRIZNGRND method for comprehensive metabolomics research on FFPE samples.

We further conducted a comparative analysis employing mxFRIZNGRND on both frozen fresh samples and FFPE samples derived from matched mouse prostate tissues. The majority of metabolites identified in fresh tissues were also detected in FFPE samples using mxFRIZNGRND (Fig. 6e, f). Consistently, the metabolite profiles obtained from both approaches were found to be highly similar (Fig. 6g). These data together demonstrate the feasibility and effectiveness of the mxFRIZNGRND method for metabolomics research on FFPE samples.

### mxFRIZNGRND results for clinical specimens

Slides from FFPE samples adjacent to the ones used for snFLARE-seq were employed for metabolite detection using the mxFRIZNGRND method. A total of 1284 lipid species were identified through lipidomics analysis, and another 489 hydrophilic metabolites were characterized using a comprehensive targeted metabolomics platform. The results of principal component analysis (PCA) showed that lipid profiles from the PZ group tended to cluster together, whereas samples from the TZ group were more dispersed, indicating more pronounced metabolic heterogeneity in transition zone-originated prostate cancer (Fig. 7a). Partial least squares discriminant analysis (PLS-DA) successfully captured the metabolic differences between any two groups, revealing distinct metabolic features among these three groups (Fig. 7b).

The differential lipids and hydrophilic metabolites were selected and categorized into four classes: high abundance exclusively in TZ samples (category I), high abundance exclusively in PTM samples (category II), low abundance exclusively in PTM samples (category III), and low abundance exclusively in PZ samples (category IV) (Fig. 7c, and Supplementary Fig. 20a–d). These signatures may be used for patient stratification and mechanistic research. Metabolite enrichment analysis of these categories revealed a consistent involvement of central carbon metabolism, choline metabolism and ABC transporters in tumor tissues (Supplementary Fig. 20e, f). Together, these findings underscore the distinct metabolic profiles associated with tumors of different origins and revealed a dormant status of lipid metabolism in PZ samples.

### Integrative analysis on the results of snFLARE-seq and mxFRIZNGRND

Results of snFLARE-seq and mxFRIZNGRND were combined together for integrative analysis. PI3K-AKT pathway and choline metabolism were identified as critical alterations for disease progression by DNB analysis, in both progression trajectories I and II (Fig. 3h, i). However, metabolites related to choline metabolism were identified as signature metabolites for category III, with low abundance in PTM samples (Supplementary Fig. 20e, f). The integrated analysis revealed that choline transporters, SLC22A and SLC44A, were upregulated in PTM samples. However, CHKB, with an increased expression, accelerated the conversion from choline to PC, resulting in a significant reduction of choline along with an increase of PC in PTM samples (Fig. 7D). The activated PI3K-AKT pathways promoted the transcription of CHKB, consistent with the DNB analysis that alterations of PI3K-AKT pathway occurred at an earlier time point than that of choline metabolism (Fig. 3h, i). As an essential component for cell membrane system, the abundant PC generated from the active PI3K-AKT-Choline-PC axis provides the material basis for cell division. To further validate the essential role of choline metabolism for prostate cancer, choline was used to treat C4-2 cells (an AR-positive cell line) and DU145 cells (an AR-negative cell line). Choline promoted cell growth in these cell lines, consistent with previous reports (Supplementary Fig. 21)[63]. For central carbon metabolism, accelerated glycolysis was found in PTM samples, as evidenced by higher abundance of glucose and upregulated expression of related enzymes (Fig. 7e). Notably, HIF1A promoted both choline/PC metabolism and glycolysis, and displayed enhanced expression in PTM samples (Fig. 7d, e). TCA cycle was suppressed in PTM samples, with reduced expression of IDH1 and fewer related metabolites detected (Fig. 7e). Surprisingly, the concentration of lactate was not increased in PTM samples, but showed a tendency to decrease. A plausible explanation could be that lactate is further used for protein post-translational modifications. For pyrimidine metabolism, enriched cytosine, uracil, and thymine were detected in PTM samples, along with related enzymes showing significant alteration, indicating a huge demand from DNA/RNA synthesis and cell proliferation (Fig. 7f). Furthermore, increased hexose ceramide and decreased ceramide, which have been reported previously to correlate with tumor progression, were also consistently detected in PTM samples by the integrative analysis (Fig. 7g)[71,72]. Together, these data revealed crucial pathways associated with the cell membrane system, DNA/RNA synthesis, energy metabolism, and signaling transduction that contribute to treatment resistance, shedding light on the exploration of potential targets for disease intervention.

### Validation of mxFRIZNGRND results with TCGA database

We further integrate our metabolomics findings with TCGA database to explore key metabolic pathways associated with disease aggressiveness (Fig. 8a). The signature metabolites from four categories were mapped onto 9 metabolic pathways (Fig. 7c). A comparative analysis of genes implicated in these metabolic pathways revealed that 118 upregulated genes and 125 downregulated genes in 7 out of 9 metabolic pathways showed significant differential expression between prostate cancer samples and adjacent normal samples (Fig. 8b, c). Survival analysis was subsequently conducted using these differentially expressed genes, leading to the identification of 42 metabolic enzymes or transporters as potential prognostic markers (Fig. 8d). Among these, 12 risk factors and 6 protective factors for prostate cancer are directly linked to the signature metabolites and then selected for further analysis (Fig. 8e). The 12 risk factors were consistently upregulated in advanced-stage prostate cancer, whereas the 6 protective factors were downregulated in advanced-stage disease (Fig. 8f, g). Notably, choline/PC metabolism, central carbon metabolism, pyrimidine metabolism, and ceramide metabolism were consistently identified as pivotal pathways for disease progression. Together, these results discovered consistent essential metabolic pathways associated with prostate cancer progression corroborated by external TCGA database.

## Discussion

Prostate cancer cells of different origins display inherent heterogeneity, and clinical interventions would exacerbate this diversity, challenging the existing, often limited approaches for disease

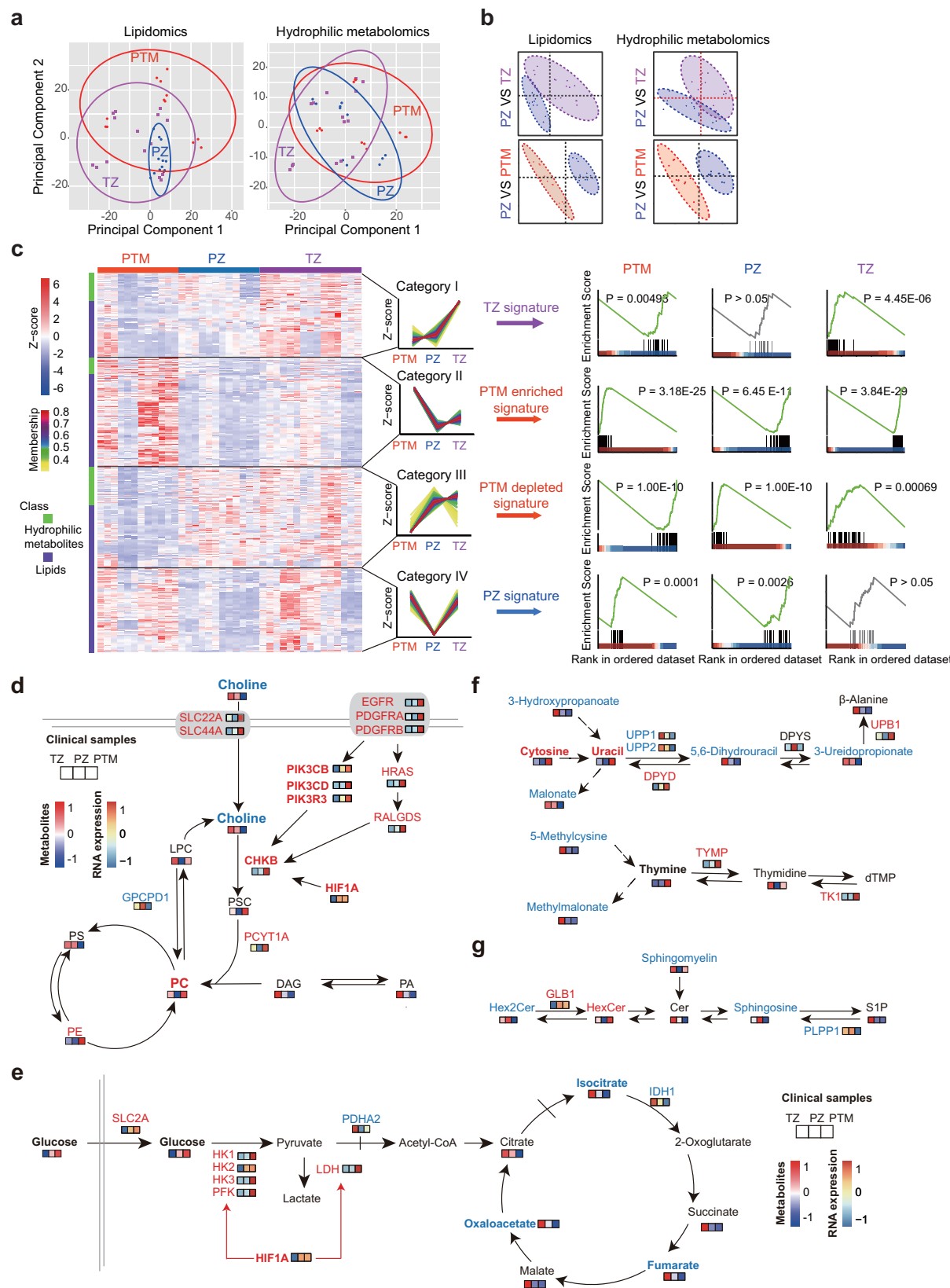

management. Multi-omics analysis of clinical specimens provides rich information on characteristics and evolution of disease heterogeneity, and lays the foundation for the development of innovative therapeutic strategies. FFPE samples are the most common preservation method for clinical specimens and are invaluable for scientific research. Here,

we developed the snFLARE-seq and mxFRIZNGRND methods for transcriptomic and metabolomic investigation on FFPE samples. With these approaches, we have elucidated the molecular characteristics of prostate cancer of different origins and highlighted the remodeling of prostate cancer and its microenvironment after hormone therapy.

**Fig. 7 | Metabolomic analysis of three cohorts. a** PCA scores plots for mxFRIZNGRND results of three cohorts. **b** Partial least-squares discriminant analysis (PLS-DA) of three cohorts. **c** Metabolic signature of different cohorts. Different lipids and hydrophilic metabolites between cohorts displayed in a heatmap were divided into four groups to distinguish PZ, TZ, and PTM metabolomics. The heatmap displays metabolites that were significantly different ($P < 0.05$, one-way ANOVA) between cohorts, which were subsequently clustered into four groups using the Mfuzz algorithm. The color scale represents the relative abundance of each metabolite compared to the TZ group (red, increased; blue, decreased). **d** Dysregulation of choline-phospholipid metabolism pathway in three cohorts, with increased phosphatidylcholine (PC) synthesis observed in PTM samples. **e** Central carbon metabolism pathway in three cohorts, showing increased glycolysis and decreased TCA cycle in PTM samples. **f** Pyrimidine metabolism pathway in three cohorts, with enriched cytosine and uracil in PTM samples. **g** Sphingolipids metabolism pathway in three cohorts, with enriched hexose ceramide (HexCer) in PTM samples. The average abundance of metabolites and gene expression in TZ, PZ, and PTM samples are marked with different colors in three consecutive boxes. Significantly changed metabolites or genes compared to TZ group with red denoting increased abundance and blue denoting decreased abundance ($P < 0.05$, one-way ANOVA). PC phosphatidylcholine, PS phosphatidylserine, PE phosphatidylethanolamine, DAG diacylglycerol, PA phosphatidic acid, PSC phosphorylcholine, LPC lysophosphatidylcholine, S1P sphingosine 1-phosphate, Cer ceramide, HexCer hexose ceramide, Hex2Cer dihexosyl-ceramide. PZ, $n = 4$, TZ, $n = 6$, and PTM, $n = 4$.

Crucial metabolic pathways and key signalings associated with disease aggressiveness have been discovered, shedding light on further disease management.

By meticulous optimization of the processes involved in sample decrosslinking, in situ reverse transcription, primer selection, and template switching, we have successfully developed the snFLARE-seq technology, with a superior balance between sample integrity and detection efficiency. Concurrently, by refining the extraction methods, solvent selection, and extraction iterations for metabolites detection in FFPE samples, we have established the mxFRIZNGRND method to facilitate the metabolomics research on FFPE samples. These methods have paved the way for characterizing prostate cancer of different origins and for deciphering disease evolution after clinical interventions in the real world.

Prostate cancer from different anatomical origins exhibits different clinical and molecular features. Our cohort study also confirms that patients with PZ prostate cancer are more susceptible to biochemical recurrence after prostatectomy than patients with TZ prostate cancer, regardless of neoadjuvant therapy receipt. This indicates that the impact of tumor origin on disease aggressiveness is consistent across different ethnicities, suggesting potential conserved molecular mechanisms underlying this clinical observation[11,13,73,74]. Our results demonstrate that the most pronounced differences between PZ cancer and TZ cancer lie in epithelial cells. Epithelial cluster with markers associated with disease aggressiveness is more enriched in PZ cancer. The microenvironment is relatively homogeneous across PZ and TZ samples. However, PZ samples have more immune-suppressive cells and aggressiveness-associated fibroblasts. These findings provide insights into the more favorable outcomes for patients with prostate cancer originating from TZ. According to our GSEA results, inhibitors targeting the IL-17 pathway may be more appropriate for PZ prostate cancer. Epithelial cells enriched in TZ samples have high expression of AMACR (Cluster 0 and 11 in Fig. 3c), thus AMACR inhibitors might be more suitable for TZ-originated prostate cancer[75]. Surprisingly, metabolomics results indicate that PZ cancer cells are more likely in a metabolism-dormant status, with uniform and diminished cellular lipid metabolism. The metabolic status of PZ samples may enable cancer cells to adapt and survive in a markedly altered metabolic milieu, thereby potentially conferring resistance to clinical interventions. The DNB analysis, designed to find the critical points in tumor evolution, revealed that PI3K-AKT signaling and choline metabolism pathways may be targeted to prevent the aggressive evolution of PZ epithelial cells.

We also find that patients with cancer areas spanning PZ and TZ have more rapid disease progression, and the corresponding tumors show a more pronounced malignant trend. PTM samples, from patients receiving adjuvant therapy for prostate cancer spanning PZ and TZ, offer a unique opportunity to elucidate the stress response of malignant prostate cancer to hormone therapy. The integrative transcriptomic and metabolomic analyzes point out that PTM samples are not at an intermediate status between PZ and TZ samples, but become more heterogeneous compared to PZ and TZ samples. The epithelial cells of PTM samples are more similar to PZ samples but show a "polarized" feature. Regarding immune cells and fibroblast cells, PTM samples exhibited a "subverted" microenvironment: cell subtypes that are less common in PZ/TZ samples are markedly enriched in PTM samples, and the abundant cell subtypes in PZ/TZ samples become obviously diminished in PTM samples. It is also important to highlight that hormone therapy selects for cells with a more active androgen-AR pathway. This is consistent with our previous findings and indicates that more thorough suppression of the androgen-AR pathway is required for further disease management[76–79]. These results indicate a profound impact of hormone therapy on the tissue environment.

The cell communication strength in PTM samples is reduced, which may suggest a more disordered and out-of-control tumor microenvironment. The communications between immune cells, particularly T cells, and epithelial cells are enhanced, indicating the feasibility of managing prostate cancer with immune therapy after hormone therapy. And T cells may still represent the most promising avenue for therapeutic breakthrough.

Given the huge impact of hormone therapy on both cancer cells and the microenvironment, our findings also stimulate discussion on the potential consequences of intensive therapy for localized prostate cancer. Our cohort study reveals that all patients receiving ADT plus neoadjuvant therapy develops castration resistant prostate cancer rapidly, particularly when compared to those in cohort 1 (Fig. 1a), who have received only radical prostatectomy without ADT. The increased heterogeneities of prostate cancer and deteriorated microenvironment after hormone therapy compromise the efficacy of clinical treatments and inevitably lead to treatment resistance[7,80]. Without approaches to distinguish patients with potentially aggressive prostate cancer from those with indolent disease, non-selective early intervention would lead to overtreatment, a concern not only for economic reasons but also because it may awaken dormant cancer cells.

While steroidogenic enzymes attract more attentions for prostate cancer management, other metabolic pathways and their associated genes are also pivotal to promote disease progression[81–83]. The integrated analysis combining mxFRIZNGRND data with snFLARE-seq results or TCGA database consistently shows that pathways such as choline/PC metabolism, central carbon metabolism, pyrimidine metabolism, and ceramide metabolism are linked to disease aggressiveness. Essential genes in these pathways, like CHKB, would be promising therapeutic targets for prostate cancer treatment beyond hormone therapy. Hypoxia signaling (HIF1A) and PI3K-AKT signaling might act as upstream regulators of these metabolic pathways, and the related inhibitors may be used to enhance the clinical efficacy of hormone therapy.

In summary, we have developed the snFLARE-seq and the mxFRIZNGRND methods to analyze the transcriptome and metabolome of prostate cancer originating from different tissue regions. Our findings demonstrate that hormone therapy substantially remodels cancer cells and their associated microenvironment. Additionally, our results provide potential targets for precise treatment of prostate cancer arising from different areas.

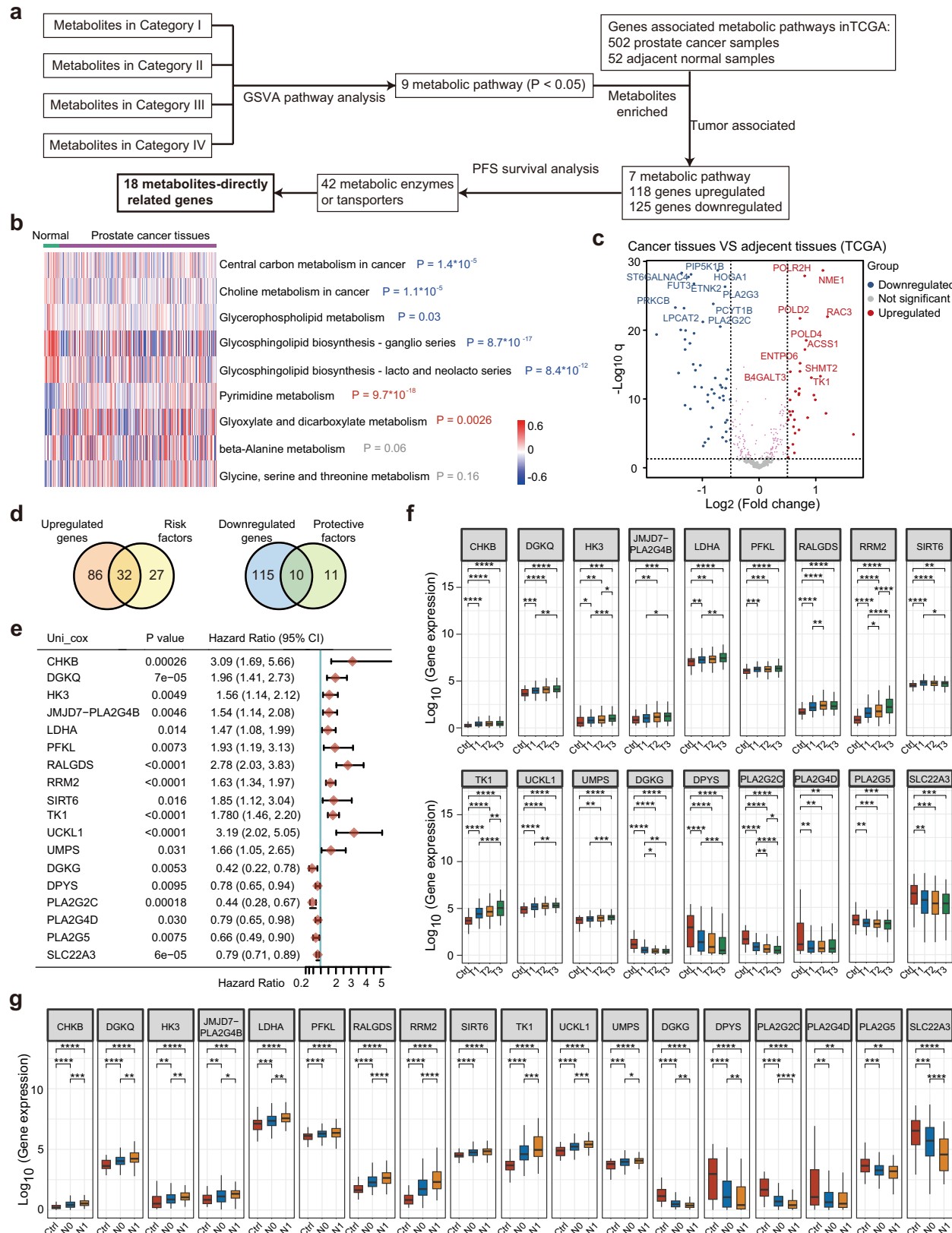

## Methods

### Clinical specimens and cell lines

Patient-derived tissues were collected in adherence to the guidelines established by the Ethics Committee of Nanjing Drum Tower Hospital (ID: 2017-044-02). Written consent was obtained from patients concerned. Only male patients were recruited for research because prostate cancer is a man-specific disease.

Patients with localized prostate cancer were selected and assigned into two groups according to the pathological results from whole-mount specimens after radical prostatectomy. Five patients

**Fig. 8 | Integrative analysis of mxFRIZNGRND results with the TCGA database.**
**a** Flow chart of the integrative analysis. **b** Gene set variation analysis (GSVA) of genes involved in the nine selected metabolism pathways in TCGA. The enrichment scores of each pathway were compared between normal ($n = 52$) and prostate cancer tissues ($n = 502$) using the two-sided Mann–Whitney U test. **c** Comparative analysis of the expression of metabolic genes involved in the metabolic pathways (adjacent tissues, 52 samples; cancer tissues, 502 samples). **d** Venn diagram showing the overlap of survival analysis and comparative analysis of gene expression. **e** Survival analysis forest map of the 18 metabolism-directly related genes in

TCGA database ($n = 498$). The hazard ratio (HR) and 95% confidence interval for each gene were calculated using univariate Cox proportional hazards regression. Error bars for hazard ratio indicate hazard ratio and 95% confidence interval (CI). **f, g** Correlations of the 18 metabolic genes with disease stages. Normal tissues, 52 samples; T1 stage, 177 samples; T2 stage, 205 samples; T3 stage, 110 samples; N0 stage, 345 samples; N1 stage, 80 samples. For box plot, median, 25th percentile, and 75th percentile are shown; whiskers extending to 1.5 X IQR (interquartile range). One-way ANOVA. $^*P < 0.05$; $^{**}P < 0.01$; $^{***}P < 0.001$; $^{****}P < 0.0001$.

with multi-focal lesions were assigned into group 1. Another four patients with residual lesions in PZ+TZ after a treatment of 6-month neoadjuvant ADT prior to radical prostatectomy were assigned to group 2. The detailed characteristics of these 9 patients are shown in Supplementary Fig. 1.

The formalin-fixed paraffin-embedded (FFPE) tissues were carefully dissected for tumor foci and adjacent tissues. A single 5-µm slice and several 20-µm slices were cut from the carved paraffin tissue. The thin slice was stained with hematoxylin-eosin (HE) to confirm the presence of both carcinoma and adjacent tissue. Subsequently, the thick slices were processed further for sequencing and metabolomics analysis. The tumor and sampling locations of each patient were graphically depicted in Supplementary Fig. 1.

LNCaP and C4-2 were purchased from the American Type Culture Collection (ATCC, Manassas, VA) and cultured in RPMI-1640 medium, supplemented with 10% FBS (ExCell Bio, China). The VCaP cell line was graciously provided by Dr. Jun Qin (SINH, Shanghai, China). All experiments with LNCaP and VCaP were performed in plates coated with poly-DL-ornithine (Sigma-Aldrich, St. Louis, MO). Authentication of cell lines was confirmed by Hybribio using STR profiling (Guangzhou, China), and mycoplasma-free status was verified using specific primers: 5′- GGGAGCAAACAGGATTAGATACCCT −3′ and 5′-TGCACCATCTGTCACTCTGTTAACCTC −3′.

### Cell nuclei preparation and pre-decrosslinking

Tissue sections of 20 µm in thickness underwent deparaffinization, rehydration, and homogenization in a pre-chilled CST lysis buffer using a Dounce homogenizer (Sigma-Aldrich, St. Louis, MO; #D8938). Subsequently, the nuclei count was estimated using a Fluorescence Cell Analyzer (either Seekgene #M002B or Countstar® Rigel S2) with AO/PI reagent after washing the samples.

The nuclear RNA was then isolated using the SeekOne® DD Single Cell Decrosslinking Kit from Seekgene (#K02101-0806). The integrity of the RNA was then assessed on the Agilent TapStation 4200. Qualified nuclei were subjected to pre-crosslink reversal at 25 °C for 30 min, followed by incubation at 80 °C for 15 min in ST(-) buffer containing 2.4 µ/ml Proteinase K (NEB, Cambridge, MA; #P8111S) and 0.1% SDS. After pre-crosslink reversal, the nuclei were gently resuspended in 1X PBS and centrifuged at $1000 \times g$ to remove any debris.

### In situ reverse transcriptions

To enhance the efficiency of primer hybridization and intracellular reverse transcription, a tailored thermal cycling protocol was implemented: fifteen cycles of annealing were performed with the following temperature profile−8 °C for 12 s, 15 °C for 45 s, 20 °C for 45 s, 30 °C for 30 s, and 42 °C for 3 min−utilizing a temperature transition rate of 1.5 °C per second. RNA present in the nuclei was reverse transcribed using a mixture of RT primers, specifically 5′P-CGTCCGTCGTTGCTCGTNNNNNNNNNNTTTTTTTTTTTTTTTTTTTT-3′ and 5′P-CGTCCGTCGTTGCTCGTNNNNNNNNNNGGGG/TTTT-3′, to initiate reverse transcription across full-length transcripts. Meanwhile, template switching was achieved using AAGCAGTGGTATCAACGCA-GAGTACATrGrG+G. After reverse transcription, the nuclei were washed twice with PBS to eliminate residual primers.

### Single-nucleus bar-coding and library construction

Single-cell RNA sequencing libraries were generated utilizing the SeekOne® Digital Droplet platform in conjunction with Barcoded Hydrogel Beads (BHBs) from the scFAST-seq[15] system (SeekGene, China; #K02101). A calculated quantity of nuclei was combined with ligation reagents and subsequently introduced into the sample wells of the SeekOne® DD Chip S3. The Barcoded Hydrogel Beads (BHBs) along with partitioning oil were dispensed into the corresponding wells of the Chip S3. The mixture containing cells, T4 DNA ligase and the bridge oligo ACGAGCAACGACGGACGACAGCAA-C3, as well as BHBs, was encapsulated into emulsion droplets using the SeekOne® Digital Droplet System.

The resulting emulsion droplets were promptly transferred into PCR tubes where they underwent a 60-min incubation at 20 °C, succeeded by a 10-min heat inactivation step at 65 °C to yield barcoded cDNA. This cDNA was then decrosslinked from the disrupted droplets and subjected to two rounds of purification. To increase the amount of Template-Switched cDNA, a secondary reverse transcription was performed, followed by a PCR amplification.

A quarter volume of the cDNA was amplified via index PCR. The indexed sequencing libraries were purified using AMPure beads and quantified employing quantitative PCR (Roche, Swiss; KK4824). The prepared libraries were then sequenced on either the NovaSeq 6000, DNBSEQ-T7, or SurfSeq 5000 platforms, utilizing PE150 read length for enhanced accuracy and comprehensive transcriptome profiling.

Nuclei were directly isolated from fresh kidney tissues in a pre-chilled NIB lysis buffer using a Dounce homogenizer, and debris was removed with density gradient centrifugation in 18% Percoll at $3000 \times g$ for 30 min. Proper number of nuclei was used to construct single nucleus RNA Sequencing library according to the commercial SeekOne® DD single-cell 3′ Transcriptome Kit (SeekGene Biosciences).

All prepared libraries were sequenced on either the Illumina NovaSeq 6000, MGIseq DNBSEQ-T7, or GeneMind SurfSeq 5000 platforms, utilizing PE150 read length for enhanced accuracy and comprehensive transcriptome profiling.

### snFLARE-seq analysis

Single-cell RNA-sequencing data were merged into a unified dataset. To ensure the data integrity, we rigorously applied Seurat's standard quality control (QC) procedures for preprocessing, which effectively filtered out non-viable cells, doublets, and other potential contaminants. This involved setting stringent thresholds for low-count cells (min.cells = 3 and min.features = 200 within the CreateSeuratObject function) and meticulously calculating and filtering mitochondrial gene expression (utilizing Seurat's PercentageFeatureSet function with criteria for nFeature_RNA > 200 & nFeature_RNA < 2500 & percent.mt <20). To mitigate batch effects, canonical correlation analysis (CCA) was performed using the Seurat package[84]. The efficacy of batch correction was visually confirmed through UMAP dimensionality reduction, both pre- and post-correction, ensuring that any potential batch effects in the snFLARE-seq data were substantially reduced. For principal component analysis (PCA), the gene expression variances among patients were graphically represented in a two-

dimensional plot, where each point corresponded to an individual patient.

Cells of the same cell type were categorized based on their origin from different prostate cancer subtypes (Peripheral/Transition Malignant, Peripheral, or Transition). Differential analysis was then performed using the FindMarkers() function in the Seurat package, followed by pathway enrichment analysis of the differentially expressed genes. To delineate the distribution patterns of clusters, odds ratios (OR) were computed to signify preferences. For each meta-cluster i and tissue j combination, a 2-by-2 contingency table was meticulously constructed, encapsulating the count of cells from meta-cluster i in tissue j, the count of cells from meta-cluster i in other tissues, the count of cells from non-i meta-clusters in tissue j, and the count of cells from non-i meta-clusters in other tissues. Fisher's exact test was then applied to this contingency table to derive the OR and its corresponding $P$ value. $P$ values were subsequently adjusted using the Benjamini–Hochberg (BH) method, as implemented in the R function p.adjust. All ORs exceeding 1.5 or falling below 0.5 were associated with adjusted $P$ values less than 1e−10. Consequently, an OR greater than 1.5 indicated a marked preference of meta-cluster i for tissue j, while an OR less than 0.5 indicated a significant aversion to tissue j[32].

## Pathway enrichment and gene set enrichment analysis

The dataset was meticulously organized according to metadata, including disease subtypes, cell type classifications from prior analyzes, sub-clusters within cell types, and clusters enriched for specific disease types. These categorical distinctions were incorporated into the metadata of the Seurat object, which was generated from the snFLARE-seq data. With these groupings, the Seurat FindMarkers function was utilized to discern differentially expressed genes among the groups, with all parameters defaulted except for the groupings. The Wilcoxon test was employed for statistical comparisons, aligning with Seurat's established workflow for single-cell data analysis. The FindMarkers function yielded a list of differentially expressed genes, complete with their respective Log2FoldChange, $P$ value, and adjusted $P$ value. $P$ value adjustments were conducted using the Bonferroni correction, calibrated to the total gene count within the dataset. Alternative correction methods were not deemed necessary, as Seurat's pre-filtering of genes based on the aforementioned criteria effectively minimized the number of tests conducted. Genes with an adjusted $P$ value of less than 0.05 were advanced to subsequent Gene Set Enrichment Analysis (GSEA).

GSEA was executed using the clusterProfiler package[85]. The gseKEGG function was engaged with parameters set to organism = "hsa" and keyType = "kegg", while other parameters remained at their default settings. Pathways of significance were identified based on the criteria of $|NES| > 1$, NOM $P$ value < 0.05, and FDR (padj) <0.25, and were subsequently visualized in the form of a ridge plot.

## Drug response estimation from single-cell expression profiles

Potential drugs for PTM epithelial cells were analyzed using the DREEP (Drug Response Estimation from single-cell Expression Profiles), a computational method that leverages publicly available pharmacogenomic screens from GDSC2, CTRP2, and PRISM and functional enrichment analysis, to predict single-cell drug sensitivity from transcriptomic data. The R package DREEP used in this study is publicly available at https://github.com/gambalab/DREEP[86]. All pre-processed datasets and accompanying metadata (including single-cell RNA-seq, bulk RNA-seq, and drug sensitivity analyses) used in the original DREEP study have been deposited in figshare under https://doi.org/10.6084/m9.figshare.23261234[86]. The drug sensitivity datasets—CTRP2 (Cancer Therapeutics Response Portal; https://portals.broadinstitute.org/ctrp/), GDSC (Genomics of Drug Sensitivity in Cancer; https://www.cancerrxgene.org), and PRISM (Profiling Relative Inhibition

Simultaneously in Mixtures; https://depmap.org/portal/prism/)—were obtained from the DepMap portal (https://depmap.org).

The batch-corrected data were first extracted using Canonical Correlation Analysis (CCA) from the Seurat object, followed by grouping cells based on prior classifications of cancer cell type and treatment condition. In this process, the DREEP workflow applied gf-icf normalization to extract the most relevant genes from each cell. Drug sensitivity genomic features (Genomic Profiles of Drug Sensitivity, GPDS) were subsequently ranked by calculating the Pearson correlation coefficient (PCC) between gene expression and drug efficacy, as measured by AUC values. Drug efficacy data were sourced from multiple datasets, including GDSC2, CTRP2, and PRISM, which provided information on cell lines where drug efficacy had been measured. Each GPDS corresponds to a ranked list of 13,849 expression-based biomarker genes, representing a vector of drug-to-gene interactions. The PCC values reflect the significance of these interactions in predicting the effects of small-molecule drugs. Lower AUC values indicate greater sensitivity to the drug, while higher AUC values suggest resistance. The GPDS list prioritizes resistance-predictive biomarkers at the top and sensitivity-predictive biomarkers at the bottom for each drug. A positive enrichment score (ES) implies that the expressed genes confer resistance to the drug, while a negative ES suggests that the expressed genes render the cells sensitive to the drug. To evaluate drug response across cell populations, the median enrichment score (ES) was calculated, alongside the proportion of cells with significantly negative ES. These metrics allow for inference of the overall drug effect on the population and the percentage of drug-sensitive cells.

## Constructing trajectories and pseudo-temporal information with Monocle3

To integrate temporal dynamics into our single-cell dataset for further analysis, we subjected epithelial cells from PTM and PZ disease subtypes to trajectory analysis using Monocle3. This software deduces the sequential gene expression patterns that cells exhibit during dynamic biological processes, circumventing the need for experimentally isolating cells into discrete states[87]. Initially, we employed the learn_graph function to delineate a developmental trajectory. We then pinpointed the earliest time point interval, characterized by the highest cell density in Cluster 9, as the root node. By applying the order_cells function, we generated pseudo-temporal information for the dataset, assigning each cell a position along the inferred trajectory.

To facilitate subsequent Differential Network Biology (DNB) analysis, it was imperative to segment the temporal information into distinct stages. This was accomplished using the choose_graph_segments function within the monocle3 package. We utilized Cluster 9 as the starting point for constructing pseudotime and Cluster 6 as one of the developmental endpoints, which is pivotal for the subsequent DNB analysis. This Monocle3-based methodology not only enabled us to construct cellular trajectories but also provided a pseudo-temporal framework, facilitating a comprehensive understanding of the developmental progression within our single-cell dataset.

## Dynamic network biomarker selection

To identify Dynamic Network Biomarkers (DNBs), we utilized the DNBr package[52]. For trajectories I and II, generated via Monocle3, the cell ordering was refined through a sliding window method, with each window encompassing 100 cells and progressing with a step size of 50 cells. Consequently, trajectory I was segmented into 18 time windows, while trajectory II was divided into 22. The DNBcompute function was then deployed to calculate DNB scores, accepting these temporal subsets as inputs. This function assessed DNB scores for multiple gene modules across various time stages, providing a comprehensive view of the network's dynamic behavior. Post-computation, the DNBfilter function was engaged to refine and select the top five modules with the highest DNB scores for each time stage. These modules, composed of

gene sets varying from a handful to several dozen, epitomized the most dynamically mutable elements within the network. Subsequently, the resultAllExtract function was applied to collate the outcomes, spotlighting modules that exhibited the most substantial alterations in DNB scores. These curated modules, along with their constituent genes, serve as a proxy for the network's most fluctuating components.

This meticulous approach, employing the DNBr package, enabled a systematic identification and extraction of Dynamic Network Biomarkers, yielding vital insights into the temporal progression of network dynamics within the biological system under scrutiny.

Subsequently, the genes from the highest-ranking modules identified by DNB were advanced to pathway enrichment analysis. This analysis was conducted using the clusterProfiler package's enrichKEGG and enrichGO functions, with parameters set to $pvalueCutoff = 0.05$ and $qvalueCutoff = 0.05$. The significant pathways were then graphically represented using bar plots, offering a visually accessible summary of the gene functions and their associated biological pathways.

## Cell communication

The consolidated Seurat object, derived from the combined single-cell dataset, was meticulously partitioned into three distinct subsets, each corresponding to a specific disease type. Subsequently, these Seurat objects were transformed into CellChat objects using the createCellChat function from the CellChat package[69], with the parameters meta set to "seurat_subset@meta.data" and group.by to "celltype". The reference database utilized was CellChatDB.human. Adhering to the established workflow, each of the three CellChat objects underwent a series of analytical processes. This included the identification of overexpressed genes and ligand-receptor pairs, the inference of intercellular communication, the exclusion of cell-cell interactions with limited cell representation in certain groups, the computation of communication outcomes for all ligand-receptor interactions associated with each signaling pathway, and the quantification of integrated communication outcomes between different cell types.

Individual visualization of each CellChat object was achieved using the custom_netVisual_heatmap and netVisual_circle functions, generating heat maps and cell-cell communication networks. These visualizations effectively depicted the quantity and intensity of communications between various cell types. Subsequently, the three CellChat objects were merged in pairs using the mergeCellChat function, with the resulting heatmaps visualized using the custom_netVisual_heatmap function. This step was crucial for elucidating the disparities in cell communication across different disease types.

To quantitatively assess the network similarity between two disease states, the computeNetSimilarityPairwise function was applied, with the type parameter set to "functional" for functional network similarity calculations. This was followed by manifold learning of the signaling networks based on their similarity, facilitated by the netEmbedding function, and classification learning of the signaling networks using the netClustering function. Ultimately, the rankNet function was employed to rank and visualize the signaling networks based on differences in information flow. This comprehensive approach served to accentuate the differential cell communication pathways between the distinct disease types, providing a nuanced understanding of the underlying biological processes.

## Signature analysis

For epithelial cell signature, the top 8 differentially expressed genes (CX3CR1, CXCL1, CAPN8, OLFM4, IGSF5, RARRES1, MUC13, SDCBP2) in cluster 6 were used as signature genes for survival analysis in TCGA and CPGEA with survminer R package. For the fibroblast cell signature, the top 7 differentially expressed genes (PAH, FAM3D, ANKRD30A, DSG3, SHC4, PGBD5, CD3E) in the enriched clusters were used as

signature genes for survival analysis in TCGA and CPGEA with survminer R package. $P$ value was calculated using the Cox proportional hazard model.

For immune cells, previously published gene signatures[39,40] were analyzed in different clusters using AddModule Score function in the Seurat package and visualized using ggplot2. $P$ value was calculated using Wilcoxon test.

## Extraction of metabolites

F-extraction: Each FFPE tissue was snap-frozen with liquid nitrogen, treated with tissuelyser (20 Hz, 1 min × 3), and mixed with 90% precooled methanol: $H_2O$ (9:1, v/v). After sonication in an ice bath for 10 min and centrifugation for 5 min (14,000 × $g$, 4 °C), the supernatant was obtained and subdivided into two portions with one reconstituted into methanol: water (1:1, v/v) for analyzing hydrophilic substances and the other reconstituted into methanol: dichloromethane (1:1, v/v) for lipidomic analysis.

H-extraction: This extraction was done as previously reported[22] with minor modification. Briefly, after extracting each FFPE sample with methanol: $H_2O$ (9:1, v/v) at 75 °C for 45 min, the supernatant was obtained and subdivided into two portions with one reconstituted into methanol: water (1:1, v/v) for analyzing hydrophilic substances and the other reconstituted into methanol: dichloromethane (1:1, v/v) for lipidomic analysis.

X-extraction: This extraction was conducted as reported[25] with minor modification. Briefly, the FFPE samples were washed with xylene followed with extraction using methanol: water (9:1). The supernatant was subdivided into two portions and dried with nitrogen gas; one was reconstituted into methanol: water (1:1) for analysis of hydrophilic metabolites and the other into methanol: dichloromethane (1:1) for lipidomic analysis.

For the method comparison studies, three independent technical replicates were conducted for each experimental condition to validate the consistency and reliability of our findings ($n = 5$). This approach allowed for statistical analysis of methodological variations and ensured the reproducibility of our comparative assessments.

mxFRIZNGRND: Each FFPE tissue was snap-frozen with liquid nitrogen, treated with tissuelyser (20 Hz, 1 min × 3) and mixed with 90% precooled methanol: $H_2O$ (9:1, v/v). After sonication in an ice bath for 10 min and centrifugation for 5 min (14,000 × $g$, 4 °C), the supernatant was obtained. The above procedures were repeated once more, and the resultant two supernatants were combined followed with drying with nitrogen gas. The extracts were mixed with 1188 μL methanol: water: MTBE (250:188:750) to obtain hydrophilic substances (lower layer) and lipids (upper layer) after centrifugation for 5 min (14,000 × $g$, 4 °C). The obtained hydrophilic substances were redissolved into methanol: water (1:1, v/v) for metabolomic analysis. The remaining tissue residues from the above extraction were further extracted with 1 mL IPA: $H_2O$ (9:1, v/v) to obtain more lipids, which were combined with lipids from methanol: water: MTBE and reconstituted into 50 μL methanol: dichloromethane (1:1, v/v) for lipidomic analysis.

The metabolite extraction protocol for frozen sections was adapted from a previously established method[22] with critical modifications to facilitate concurrent metabolomic and lipidomic analysis and comparison with formalin-fixed paraffin-embedded (FFPE) sections. Briefly, following solvent extraction and drying under a stream of nitrogen, the dried extracts were reconstituted in 1188 μL of a methanol: water: MTBE (250:188:750, v/v/v) mixture to induce phase separation. After centrifugation (14,000 × $g$, 5 min, 4 °C), the upper lipid-containing layer and the lower hydrophilic metabolite-containing layer were collected separately. The upper layer was dried under nitrogen and reconstituted in 50 μL of methanol: dichloromethane (1:1, v/v) for lipidomic analysis, while the lower layer was similarly dried

and reconstituted in methanol:water (1:1, v/v) for metabolomic analysis ($n = 3$).

## LC-MS method

Hydrophilic metabolite analysis was performed using a Shimadzu Nexera UHPLC system coupled to a Sciex 6500 QTrap mass spectrometer, with chromatographic separation on a Waters HSS T3 column (1.8 μm, 2.1 × 100 mm) maintained at 50 °C. An injection volume of 1 μL was applied in splitless mode, and the mobile phase consisted of water (A) and acetonitrile (B), both containing 0.1% formic acid, delivered at a flow rate of 0.35 mL/min under the following gradient: 5% B (0–1 min), increased linearly to 100% B (1–11 min), held at 100% B (11–13 min), and then re-equilibrated at 5% B (13.1–16 min). Mass spectrometric detection was carried out in Multiple Reaction Monitoring (MRM) mode with electrospray ionization, using separate acquisitions for positive and negative polarities. Ion source parameters were set as follows: temperature 350 °C, curtain gas 30 psi, collision gas 55 psi, and ion spray voltage at 5000 V (positive)/−4500 V (negative). The MRM transitions, along with optimized declustering potentials and collision energies, were established based on our in-house library developed from authentic standards (Supplementary Data 2). Representative total ion chromatograms (TICs) for the methods are provided in Supplementary Fig. S22.

Lipidomic analysis was performed on a Sciex 6500 QTrap LC–MS system, using three different columns—Agilent Zorbax Eclipse Plus C18 (2.1 × 100 mm, 1.8 μm) and Waters BEH HILIC (2.1 × 100 mm, 1.7 μm) with 1 μL injections, and a Phenomenex Kinetex C18 (2.1 × 100 mm, 2.6 μm) with 2 μL injections. The mobile phases consisted of water/methanol/acetonitrile (1:1:1, v/v/v) with 0.1% formic acid and 1 mM ammonium acetate (phase A) and isopropanol/acetonitrile (90:10, v/v) with 0.05% formic acid and 10 mM ammonium acetate (phase B) for the EC18 column; water/acetonitrile (5:95, v/v) with 10 mM ammonium acetate (phase A) and water/acetonitrile (1:1, v/v) with 10 mM ammonium acetate (phase B) for the HILIC column; and water/methanol/acetonitrile (1:1:1, v/v/v) with 7 mM ammonium acetate (phase A) and isopropanol with 7 mM ammonium acetate (phase B) for the C18 column. The gradient elution conditions were as described in our previously published work[88–90]. Mass spectrometric detection was conducted in MRM mode with electrospray ionization using the following parameters: ion source temperature 350 °C, curtain gas 40 psi, collision gas 55 psi, ion spray voltage ±4500 V, declustering potential 70 V, and entrance potential 10 V. The MRM transitions were sourced from an in-house library established using a Zeno TOF 7600 system, and absolute quantification was achieved via stable isotope-labeled internal standards using the formula: Concentration = (Analyte Peak Area/Internal Standard Peak Area) × Concentration of Internal Standard. The standard list was listed in Supplementary Data 3. Representative TICs for the methods are provided in Supplementary Fig. S22.

Data acquisition and analysis were done using Sciex software Analyst (Version 1.7) and SCIEX OS (Version 3.3). Metabolites with |$\log_2$(Fold Change)| > 0.5 and adjusted $p$-values < 0.05 (Storey method) from two-side Wilcox tests were identified as statistically significant high/low-abundance metabolites.

## Metabolomic and lipidomic profiling of patient-derived samples

Metabolomic and lipidomic profiling was performed on the 14 FFPE clinical specimens detailed in Supplementary Fig. 1. To account for potential technical variability during sample processing and analysis, each biological specimen was analyzed with three technical replicates ($n = 3$). These replicates were derived from consecutive tissue sections of the same FFPE block, ensuring they represented the same biological source while allowing for assessment of analytical precision. The sample preparation was carried out as described in the "Extraction of

metabolites" section using the mxFRIZNGRND method. The LC-MS method was carried out as described in the "LC-MS method" section. The area of each FFPE tissue section was measured using ImageJ software[91], and metabolite levels for each sample were normalized to this value (area normalization) to minimize variations in sample size. For data preprocessing, metabolic features with more than 20% missing values were excluded from subsequent analysis, while the remaining missing values were imputed using a random forest approach. Quality control (QC) samples were injected every eighth analytical run to monitor instrument stability. Only metabolic features exhibiting a coefficient of variation (CV) of less than 30% in the QC samples were retained for final data analysis. All processed raw data are available in Supplementary Data 4 and 5.

## Integration analysis of snFLARE-seq and mxFRIZNGRND results

Initially, significantly enriched metabolic pathways were identified by using "ClusterProfiler" R package and pathway information from KEGG. The enrichment analysis of lipidome was conducted by LION. Statistical significance was defined as adjusted $p$-values < 0.05 (with multiple testing correction using the Storey method). Subsequently, enzymes directly associated with the metabolic pathways as per KEGG were selected for integration and the snFLARE-seq data were used to identify enzymes that exhibited significant differences among PZ, TZ, and PTM groups through one-way ANOVA analysis. Metabolites and enzymes that were significantly enriched in the PTM group were marked in red, those significantly depleted in the PTM group were marked in blue, and those with no significant changes were marked in black.

## Statistics & reproducibility

One-way ANOVA and Wilcoxon test were employed to assess the differences between groups. The Benjamini–Hochberg procedure was used for $P$ value correction in multiple comparisons. A $P$ value of less than 0.05 was used to determine statistical significance. *, ** and *** denoted $P < 0.05$, $P < 0.01$ and $P < 0.001$, respectively. All analyzes were performed using R version 4.2.2 software. GSVA was performed by GSVA package[92]. PCA and PLSDA were conducted by ropls package[93]. Cluster analysis was conducted by mfuzz package[94]. Enrichment of metabolic pathways was conducted with the ClusterProfiler package[95]. Data represent the median ± interquartile range, unless indicated otherwise. No statistical method was used to predetermine sample size.

## Study approval

Patient-derived tissues were collected in adherence to the guidelines established by the Ethics Committee of Nanjing Drum Tower Hospital (ID: 2017-044-02). Informed consent was obtained from all patients concerned.

## Reporting summary

Further information on research design is available in the Nature Portfolio Reporting Summary linked to this article.

## Data availability

The metabolomic data generated in this study have been deposited in the OMIX database under accession code OMIX013309 and OMIX013310. The transcriptomic data generated in this study have been deposited in the Gene Expression Omnibus (GEO) database at the National Center for Biotechnology Information (NCBI) under accession number GSE317733. These data are also already publicly available in the National Omics Data Encyclopedia (https://www.biosino.org/node; accession OEP 00005630). Datasets derived from TCGA are available at the cBioPortal website (www.cbioportal.org/). CPGEA sequencing data are available in the Biological Project Library (PRJCA001124) (https://ngdc.cncb.ac.cn/bioproject/browse/PRJCA001124)[41]. GRCh38

human reference genome was download from 10X genomics (https://support.10xgenomics.com/single-cell-gene expression/software/downloads). The publicly available single-cell RNA sequencing data used in this study and related signatures for M2 macrophage, Antigen_Presenting, Exhaustion T cells, and Treg are accessible from the NCBI Gene Expression Omnibus database GSE181294[39]. The prostate cancer publicly available data used in this study are available in the OMIX database under accession code OMIX008930[40]. The publicly available bulk RNA-seq for prostate cancer (GSE21034) are from the NCBI Gene Expression Omnibus database (https://www.ncbi.nlm.nih.gov/geo/query/acc.cgi?acc=GSE21034)[96]. Spectra data for metabolites is shown in Supplementary Data 6. Source data are provided with this paper.

## Code availability

All scripts developed in this study are openly available under the MIT License. The source code associated with this paper can be accessed from the public GitHub repository FFPE. The code has also been archived in Zenodo under the https://doi.org/10.5281/zenodo.17779236[97]. The repository contains R scripts for transcriptomic and metabolomic analyses based on the snFLARE-seq and mrFRIGID pipelines. Part of the figures were generated using ChemDraw 18.0 and the Figdraw platform (www.figdraw.com).

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

## Acknowledgements

We thank the staff members (Pengyu Wang, et al.) of Bio-Med Big Data Center /Shanghai Institute of Nutrition and Health (SINH) for providing technical support and assistance in data collection and analysis. We acknowledge financial supports from the National Key R&D program of China (2023YFC3402700, Z. Li), the Strategic Priority Research Program of the Chinese Academy of Sciences (XDB0990000, Z. Li), National Natural Science Foundation of China (82430090, Z. Li; 82172639, X. Qiu; 82372738, S. Huang; 92157101, Z. Li; 82504485, Q. Chen; T2341007, T2350003, 31821002, and 12131020, L. Chen), National Key R&D program of China (2023YFC3404700, Z. Li; 2022YFC3400700, H. Tang; 2022YFA0806400, Q. Wang; 2025YFF1207900, 2025YFC3409300, L. Chen), Natural Science Foundation of Shanghai (23ZR1469800, Z. Li; 23JS1401300, L. Chen), Natural Science Foundation of Jiangsu province (BK20220777; M. Che), the Research Funds of Hangzhou Institute for Advanced Study, UCAS (2025HIAS-ZL006, Z. Li), CAS Youth Inter-disciplinary Team (Z. Li), State Key Laboratory of Drug Research (SKLDR-2024-KF-03, Z. Li), Tianfu Jincheng Laboratory (TFJCPI20260001, L. Chen) and JST Moonshot R&D (JPMJMS2021, L. Chen). Part of the figures were generated using ChemDraw 18.0 and the Figdraw platform (www.figdraw.com).

## Author contributions

Z. Li, X. Qiu, H. Guo, and D. He designed the study; H. Hu, Q. Chen, X. Mu, Q. Wang and H. Tang developed mxFRIZNGRND, acquired and analyzed the metabolism data; S. Jiao, G. Sang, and X. Zhi developed snFLARE-seq; Y. Cheng, S. Peng, H. Guo, X. Qiu, and S. Huang performed clinical retrospective analysis and collected specimens; K. Xiao, Y. Hao, Y. Cai, Y. Zhang, Y. Chang, Z. Liu, X. Yan, Q. Ye, Y. Yang, L. Chen, and Z. Li analyzed the snFLARE-seq results. S. Jiao, H. Hu, K. Xiao, M. Che, H. Tang and Z. Li drafted the manuscript. All authors discussed the results and revised the manuscript.

## Competing interests

Shaozhuo Jiao is a co-founder of SeekGene Biosciences Inc. Guoqin Sang, Xiaoling Zhi, and Shaozhuo Jiao are listed as inventors on a patent application pertaining to this work. The remaining authors declare no competing interests.

## Additional information

**Dongyin He**[1,2,12], **Haoran Hu** ®[3,12], **Kai Xiao** ®[2,12], **Yuhang Zhang**[2,12], **Yongbing Cheng**[4,5,12], **Shaozhuo Jiao** ®[6,12], **Yimei Hao**[7,12], **Yuanyuan Cai**[2], **Ziqun Liu**[2], **Xinran Yan**[2], **Qinsheng Chen**[3], **Xiyan Mu**[3], **Qi Wang**[3], **Shan Peng** ®[8], **Guoqin Sang**[6], **Xiaoling Zhi**[6], **Yanxia Chang**[6], **Qing Ye**[9], **Yuyao Yang**[9], **Meixia Che**[10], **Shengsong Huang** ®[1] ✉, **Hongqian Guo** ®[4,5] ✉, **Luonan Chen** ®[2,9,11] ✉, **Huiru Tang** ®[3] ✉, **Xuefeng Qiu**[4,5] ✉ & **Zhenfei Li** ®[2,9] ✉

[1]Department of Urology, Tongji Hospital, School of Medicine, Tongji University, Shanghai, China. [2]Key Laboratory of Multi-Cell Systems, Shanghai Institute of Biochemistry and Cell Biology, Center for Excellence in Molecular Cell Science, Chinese Academy of Sciences, University of Chinese Academy of Sciences, Shanghai, China. [3]State Key Laboratory of Genetics and Development of Complex Phenotypes, School of Life Sciences, Human Phenome Institute, Zhangjiang Fudan International Innovation Center, Metabonomics and Systems Biology Laboratory at Shanghai International Center for Molecular Phenomics, Zhongshan Hospital, Fudan University, Shanghai, China. [4]Department of Urology, Nanjing Drum Tower Hospital, Affiliated Hospital of Medical School, Nanjing University, Nanjing, China. [5]Institute of Urology, Nanjing University, Nanjing, China. [6]SeekGene BioSciences Co. Ltd, Beijing, China. [7]Department of Laboratory Diagnostics, Changhai Hospital, Navy Military Medical University, Shanghai, China. [8]Department of Pathology, Nanjing Drum Tower Hospital, Affiliated Hospital of Medical School, Nanjing University, Nanjing, China. [9]Key Laboratory of Systems Health Science of Zhejiang Province, School of Life Science, Hangzhou Institute for Advanced Study, University of Chinese Academy of Sciences, Hangzhou, China. [10]State Key Laboratory of Pharmaceutical Biotechnology, Chemistry and Biomedicine Innovation Center (ChemBIC), School of Life Sciences, Nanjing University, Nanjing, China. [11]School of Mathematical Sciences and School of AI, Shanghai Jiao Tong University, Shanghai, China. [12]These authors contributed equally: Dongyin He, Haoran Hu, Kai Xiao, Yuhang Zhang, Yongbing Cheng, Shaozhuo Jiao, Yimei Hao. ✉e-mail: hssfline@tongji.edu.cn; dr.ghq@nju.edu.cn; lnchen@sjtu.edu.cn; huiru_tang@fudan.edu.cn; xuefeng_qiu@nju.edu.cn; zhenfei.li@sibcb.ac.cn

