## [Transparent Peer Review file · Nature Communications]

Analysis of the transcriptomic and metabolomic landscape of prostate cancer with different anatomical origins using snFLARE-seq and mrFRIZNGRND

Corresponding Author: Professor Zhenfei Li

Version 0:

Reviewer comments:

Reviewer #1

(Remarks to the Author)

In the present work, the authors have developed innovative methodologies, snFLARE-seq and mrFRIGID, to investigate the transcriptomic and metabolomic landscapes of prostate cancer using formalin-fixed paraffin-embedded specimens. Their findings reveal significant molecular differences between cancer cells originating from the peripheral zone and transition zone, particularly in response to hormone therapy, which reshapes both cancer cells and their microenvironment. The study provides valuable insights into disease heterogeneity and potential therapeutic targets for precision medicine. However, in my opinion, in its current form, some minor changes are required for its publication. Following some observations and recommendations that I consider that should be taken into account:

- How the differential expressed gene analysis was conducted? I believe that would be extremely interesting to investigate what genes have contributed the most to the separation between clusters in UMAP. To this aim some useful techniques are gradient boosting or SHAP analysis.

- Differentially expressed genes could also be utilized to create a gene set for comparison with other tumors, allowing to determine whether the patterns observed in this study can be extrapolated to different tumor types.

- The paper describes multiple extraction methods (F-method, H-method, X-method, and mrFRIGID) for isolating metabolites. While this diversity can provide a comprehensive view of the metabolome, it may introduce variability in the results. How this fact has been taken into account in this study?

- What criteria were used to determine whether a metabolite is up-regulated or down-regulated? Additionally, I believe it is more appropriate to refer to metabolites as having high/low or increased/decreased abundance instead of up/down-regulated.

- A more detailed and comprehensive explanation on how the integration between snFLARE-seq and mrFRIGID456 was carried out would improve the understanding of this part of the analysis

Reviewer #2

(Remarks to the Author)

In this study entitled "snFLARE-seq and mrFRIGID for the transcriptomic and metabolomic landscape of prostate cancer with different anatomical origins" He et al. investigate the molecular differences between prostate tumors sectioned from the peripheral zone (PZ) and the transition zone (TZ). Their research question is of interest to the field and merits attention however the manuscript could be better focused on the important questions and at the moment it is very dense and not so easy to follow.

The number of prostatectomy samples included in the study – 270 treatment-naïve and 218 neoadjuvant ADT-treated samples – is impressive and they would be enough to address the question at hand.

However, only 14 samples are analyzed for snRNA-seq and metabolites profiling and this limits the impact of the study.

The authors report also two novel laboratory protocols, snFLARE-seq and mrFRIGID, that have been used in analyzing these tumor samples.

These ambitious efforts to prove the viability of these two methods, in addition to addressing the biological question of PZ vs TZ characteristics, results in a compromise that provides unfortunately somewhat limited evidence for proving either of the efforts worthy.

Below are our major comments:

(1) One of the findings presented in the manuscript, as stated on lines 139-140, is that "cancers spanning PZ and TZ are more aggressive to resist clinical treatment." Considering that survival analysis with biochemical recurrence as endpoint is the only analysis performed on the complete set of 270 + 218 samples, the presentation of these numbers at the start of results misrepresents the amount of samples used for molecular analyses and should be clarified throughout the manuscript, and to our opinion do not provide sufficient evidence for the conclusions.

We have some remarks on the comparisons that have been used to arrive to the conclusion mentioned above:

- Presence of 3+3 GS tumors, notoriously indolent is low in the PZ+TZ, while the same group is enriched with T4 stage PCa.
- While perhaps not explicitly stated before, it is widely recognised that larger tumors, i.e. tumor spanning multiple anatomical regions have more cancer cells and are consequently "more aggressive" and predisposed to be more treatment-resistant than smaller tumors.
- Genetics status of the tumors, which can also determine a more aggressive tumor phenotype is not taken into account: e.g. PTEN loss, P53 and ERG-fusion status, are just few of the obvious alterations associated with aggressive phenotypes and these should be accounted for in the analysis.
- Biochemical recurrence is a surrogate endpoint and in the analysis the tumor are characterised for their index lesion only. As prostate cancer can also be multifocal, lymph nodes molecular characterisation might offer a more accurate endpoint to delineate "zone-related tumor heterogeneity and its correlation with disease aggressiveness"

(2) Effective utilisation of FFPE preserved biological material is a major hurdle in not only prostate cancer research, but in every context where FFPE-preserved material is used. Novel, improved methods for extracting RNA and metabolites would have be in high demand. With regards to RNA, the authors reference existing alternatives snPATHO-seq and snRandom-seq in their introduction, but only provide a very brief benchmark of their snFLARE-seq method against snRandom-seq. Rather, the authors use a variety of tissues to conclude that "All tissue type samples achieved a median genes per cell count of above 900, indicating the robustness of snFLARE-seq across various malignancies" (205-207). This result provides no significant improvement to snPATHO-seq (PMID:39414943), although snFLARE-seq is still worthy of further development for investigation of archival material. As per the evidence shown, more benchmarks are needed, to demonstrate which method could be used to obtain better results on archival material. We would therefore suggest reworking the title of the study, focusing on the biological observations derived from the investigation of the different anatomical compartments of the prostate, or on the description and benchmark of the novel methods snFLARE-seq and mrFRIGID.

(3) The authors report snRNA-seq results from 101,729 cells extracted from 14 samples and 9 patients (5 treatment-naïve, 4 neoadjuvant-treated). The results recapitulate previously described epithelial cell subtypes, including tumor, luminal, basal, club, hillock, and neuroendocrine cells. These data are novel, considering that previous prostate scRNA-seq datasets have been produced exclusively from the fresh or fresh frozen tissue. However, the consequent analysis is hampered by the small number of patients and samples, especially considering the wide range of conditions covered (TZ, PZ, treatment-naïve, neoadjuvant ADT-treated). The authors present a finding that club cells are enriched post-treatment. Further analysis regarding the epithelial cells are limited to presenting enriched gene ontology pathways with little to limited interpretation of the underlying biology.

(4) The enrichment of exhausted T cells, regulatory T cells, and M2-polarized macrophages post-treatment is an interesting and novel observation. Unfortunately this analysis is hampered by the low number of samples from which the single data has been generated from, making it difficult to generalise this result. In this case, datamining currently available dataset may corroborate the data, however, would still be limited in consideration of the conditions that should be covered (TZ, PZ, treatment-naïve, neoadjuvant ADT-treated) to provide some additional evidence.

Minor comments

(1) Placement of figures and figure legends significantly worsens the readability of the manuscript.

(2) mrFRIGID method is another important novel, potentially improved method for extracting valuable information from archival material. The reviewers did not accurately review it since we feel we did not have enough expertise. We however feel that the name of this method could be changed to avoid contextual misinterpretations.

Reviewer #3

(Remarks to the Author)

Reviewer #5

(Remarks to the Author)

This study presents a new multi-omics approach to investigate the heterogeneity of prostate cancer originating from different anatomical zones and its evolution under hormone therapy. The authors developed two novel methodologies, snFLARE-seq for single-nucleus transcriptomics and mrFRIGID for metabolomics, tailored for FFPE samples, addressing longstanding technical challenges in analyzing archived clinical specimens. By integrating these techniques with clinical data, the work reveals distinct transcriptomic and metabolic profiles between zone-specific tumors, identifies therapy-induced remodeling of epithelial cells and immune microenvironments, and links key pathways such as PI3K-AKT and choline metabolism to disease aggressiveness. The cohort analyses provide clinically relevant insights, such as poorer outcomes for tumors spanning multiple zones and the role of androgen receptor signaling in treatment resistance. However, limitations in sample diversity, mechanistic depth, and technical transparency hinder broader applicability.

1. The study relies on a small cohort with only nine patients for omics analysis and 270 for clinical data, raising concerns about statistical power and generalizability. All patients appear to be East Asian, which introduces potential bias given known ethnic disparities in prostate cancer biology. For example, African ancestry populations exhibit higher mutational burdens and distinct molecular subtypes not represented here. Expanding cohorts to include diverse ethnicities and larger sample sizes would enhance robustness. Additionally, the exclusion of patients without postoperative PSA decline may omit atypical cases, limiting exploration of outlier biology. Independent validation in multiethnic cohorts is critical to confirm the universality of zone-specific signatures.

2. While the study identifies PI3K-AKT and choline metabolism as critical pathways, functional validation is limited to computational predictions and basic cell proliferation assays. The proposed role of hyperforin in targeting IL-17 pathways lacks in vivo evidence or detailed mechanistic studies. Orthogonal approaches such as CRISPR-based gene editing, organoid models, or murine xenografts are needed to confirm causality. Similarly, metabolic dependencies like CHKB-mediated phosphatidylcholine synthesis require isotope tracing or pharmacological inhibition to establish their necessity in tumor progression. Without such validation, the therapeutic potential of these targets remains speculative and risks misdirecting future research.

3. The analysis of immune cell subtypes lacks spatial context, a critical gap given the importance of localized cell interactions in therapy resistance. Single-cell RNA-seq data alone cannot resolve spatial relationships between epithelial cells, fibroblasts, and immune populations. Techniques like spatial transcriptomics or multiplex immunofluorescence would clarify whether enriched Tregs or M2 macrophages localize to specific tumor niches. Furthermore, the absence of functional immune assays such as T-cell cytotoxicity or macrophage phagocytosis limits the ability to corroborate transcriptomic signatures of exhaustion or suppression.

4. The study does not fully address technical biases inherent to snFLARE-seq, despite its dual use of random and oligo-dT primers. While the method demonstrates improved gene detection in FFPE samples, its performance on non-FFPE samples such as cell lines or fresh tissues remains unclear. For instance, how does snFLARE-seq compare to mainstream 3' scRNA-seq platforms like 10X Genomics in detecting exon versus intronic reads or integrating with existing datasets? The proportion of intronic reads, which may reflect pre-mRNA capture, is not quantified, potentially complicating comparisons with polyA-selected datasets. Additionally, the method's bias toward longer RNA fragments or genes due to FFPE-induced degradation is not discussed. Systematic benchmarking against gold-standard fresh-tissue protocols and detailed reporting of read distribution across gene bodies would help users assess compatibility with their research goals.

Version 1:

Reviewer comments:

Reviewer #1

(Remarks to the Author)

I would like to thank the authors for their thorough and thoughtful revisions. They have comprehensively addressed/justified all the points raised in my initial review, and the manuscript is now significantly strengthened. I believe it is in a suitable state for publication.

The new analysis testing the derived gene signatures against the TCGA database was a particularly valuable addition. The finding that the cell communication signature may serve as a robust prognostic marker across multiple cancers is an important result that substantially increases the impact of the work.

Furthermore, the authors have successfully resolved all other points that required clarification. Their provided rationale for their choice of bioinformatic methods is scientifically sound, and potential concerns about methodological variability have been nullified. The methodology is now more transparent with the inclusion of specific statistical criteria and a clear explanation of the multi-omics integration.

In summary, the manuscript has been improved considerably based on the review feedback. The conclusions are now robustly supported, and I have no further reservations. I recommend this manuscript for publication.

Reviewer #2

(Remarks to the Author)

The authors have sufficiently addressed the reviewers' comments. However these reviewers feel that the method mrFRIGID, should be renamed : e.g. mxFRIZNGRND (Metabolites extract for FFPE samples using freezing and Grinding).

Some minor spell and grammatical check still required.

Reviewer #3

(Remarks to the Author)

Reviewer #6

(Remarks to the Author)

This study developed innovative snFLARE-seq and mrFRIGID methodologies, addressing longstanding technical challenges in analyzing archived clinical specimens, to investigate prostate cancer transcriptomics and metabolomics using FFPE specimens. The integrative analysis identified four metabolic pathways and 64 genes associated with disease aggressiveness. This paper demonstrates strong innovation and a comprehensive, systematic framework. The authors have adequately addressed previous concerns. It is recommended to be accepted after minor revision.

Minor suggestion:

Please discuss the established finding that peripheral zone-originating prostate cancer shows greater aggressiveness in both Caucasian and African populations (could be added to Introduction/Discussion with citations).

Reviewer #1 (Remarks to the Author)

In the present work, the authors have developed innovative methodologies, snFLARE-seq and mrFRIGID, to investigate the transcriptomic and metabolomic landscapes of prostate cancer using formalin-fixed paraffin-embedded specimens. Their findings reveal significant molecular differences between cancer cells originating from the peripheral zone and transition zone, particularly in response to hormone therapy, which reshapes both cancer cells and their microenvironment. The study provides valuable insights into disease heterogeneity and potential therapeutic targets for precision medicine. However, in my opinion, in its current form, some minor changes are required for its publication. Following some observations and recommendations that I consider that should be taken into account:

- How the differential expressed gene analysis was conducted? I believe that would be extremely interesting to investigate what genes have contributed the most to the separation between clusters in UMAP. To this aim some useful techniques are gradient boosting or SHAP analysis.

Reply:

Thank you for this question.

We first identified distinct cell types through clustering. Cells of the same cell type were categorized based on their origin from different prostate cancer subtypes (Peripheral/Transition Malignant, Peripheral, Transition). Differential analysis was then performed using the FindMarkers() function in the Seurat package, followed by pathway enrichment analysis of the differentially expressed genes (**Fig. 2b**).

Next, we further subdivided major cell types, such as epithelial cells, into subclusters (**Fig. 3a**). Using the Odds Ratio (OR) enrichment algorithm (1), we determined the prostate cancer subtypes in which each subcluster was enriched. The FindMarkers() function was then used to analyze the differential genes in subclusters that were either enriched or depleted in malignant types. For example, in epithelial cells, subcluster 6 was enriched in the PTM group, whereas subclusters 0 and 11 were depleted. The FindMarkers() function revealed a set of differential genes between subcluster 6 and subclusters 0 and 11, forming a signature. This signature was validated in the TCGA and CPGEA databases, where it successfully stratified patients with aggressive prostate cancer.

We hypothesize that cancer subtypes exhibit certain evolutionary characteristics. Rather than emphasizing individual gene contributions to cluster separation, we focused on dynamic fluctuations within gene networks during cancer evolution. To this end, we reconstructed evolutionary trajectories for cancer cell clusters (**Extended Data Fig. 10b**) and employed the Dynamic Network Biomarker (DNB) algorithm developed by Chen Lab to identify genes exhibiting the most significant contributions to network dynamics along these trajectories, particularly during critical state transitions. Our DNB algorithm has previously demonstrated superior performance in different disease models (2-4). In epithelial cells, we established two trajectories and identified two pivotal transition points, respectively. The most dynamically significant gene modules at each juncture were characterized and subsequently subjected to pathway enrichment analysis (**Fig. 3 h and i**). We have added these related information in the revised manuscript, line 735-739.

While gradient boosting and SHAP (SHapley Additive exPlanations) analysis have been widely adopted for feature importance analysis in disease research (5-7), these approaches were not implemented in our current workflow. Gradient boosting constructs robust predictive models by iteratively combining weak learners (typically decision trees), enabling identification of genes most influential for cluster discrimination. SHAP values provide a unified framework for interpreting machine learning outputs by quantifying each feature's contribution to individual predictions. We are interested in incorporating these advanced algorithms into our analytical platform to enhance the comprehensiveness of future investigations.

- Differentially expressed genes could also be utilized to create a gene set for comparison with other tumors, allowing to determine whether the patterns observed in this study can be extrapolated to different tumor types.

Reply:

Thank you for this inspiring question. We have created signatures for epithelial cells (**Fig. 3e**), fibroblast cells (**Fig. 4k**), and the cell communication pathway (**Fig. 5f**). Subsequently, we performed the suggested analysis using the TCGA database. The epithelial cell signature successfully stratified high-risk patients in 6 out of 9 different

cancer types (**Revised figure 1a**). In contrast, the fibroblast signature failed to serve as a reliable marker for patient stratification across all 9 tested cancer types (**Revised figure 1b**). Intriguingly, the cell communication signature distinguished high-risk patients in every single one of the different cancer types (**Revised figure 1c**). These findings are truly thought-provoking. They may indicate that cancer cells, originating from epithelial cells, share a relatively high degree of similarity. Meanwhile, fibroblast cells in the microenvironment are highly variable. However, the essential signaling pathways supporting the survival of cancer cells within the microenvironment remain consistent, providing potential common therapy targets across different cancer types. We have included part of these results in the revised manuscript (**Figure 5g**).

- The paper describes multiple extraction methods (F-method, H-method, X-method, and mrFRIGID) for isolating metabolites. While this diversity can provide a comprehensive view of the metabolome, it may introduce variability in the results. How this fact has been taken into account in this study?

Reply:

Thank you for this question. The H-method and X-method are established metabolite extraction techniques derived from previous literatures, and we have developed the F-method as a further optimized approach. The mrFRIGID method, building upon the F-method, incorporates enhancements in the number of extraction steps and the separation of hydrophilic-lipophilic layers. Notably, the H-method and X-method are not integrated into our mrFRIGID method. To avoid confusion with the mrFRIGID method, we have renamed F-method, H-method, and X-method as **F-extraction**, **H-extraction**, and **X-extraction**, respectively. The mrFRIGID method, utilized for analyzing PZ, TZ, and PTM samples, exhibits limited variability and ensures reliable and consistent comparison of results.

- What criteria were used to determine whether a metabolite is up-regulated or down-regulated? Additionally, I believe it is more appropriate to refer to metabolites as having high/low or increased/decreased abundance instead of up/down-regulated.

Reply:

Thank you for this question. The abundance of metabolites was compared among different groups (PZ, TZ, and PTM). Metabolites with $|\log_2(\text{Fold Change})| > 0.5$ and adjusted p-values < 0.05 (Storey method) from t-tests were identified as statistically significant high/low-abundance metabolites. We have included these information in the Method part, line 932-934, in the revised manuscript. In response to the reviewer's suggestion, we have revised the terminology and now refer to these metabolites as "high/low" or "increased/decreased" in abundance instead of "up-regulated" or "down-regulated".

- A more detailed and comprehensive explanation on how the integration between snFLARE-seq and mrFRIGID456 was carried out would improve the understanding of this part of the analysis

Reply:

Thank you for this question. A systematic approach with well-defined steps has been employed to integrate snFLARE-seq and mrFRIGID data:

1. Identification of enriched metabolic pathways: Significantly enriched metabolic pathways were identified by conducting pathway enrichment analysis using “ClusterProfiler” R package and pathway information from KEGG. The enrichment analysis of lipidome were conducted by LION platform. Statistical significance was defined as adjusted p-values < 0.05 (with multiple testing correction using the Storey method). These pathways are presented in Extended Data Figure 19 e and f.
2. Analysis of pathway-associated metabolites and enzymes: Metabolites and their associated enzymes related to these significantly enriched pathways were analyzed. Enzyme-pathway relationships were retrieved from the KEGG database, retaining only enzymes with direct catalytic roles in the identified pathways. Metabolic pathways containing fewer than three annotated metabolites that were too distantly related (e.g., belonging to unrelated classes or functions) were excluded for further analysis.
3. Multi-group comparison of metabolite and enzyme dynamics: snFLARE-seq data were utilized to identify metabolites and enzymes that exhibited significant differences among PZ, TZ, and PTM groups through one-way ANOVA analysis. A heatmap was generated to display the levels of metabolites and enzymes in the PZ, TZ, and PTM groups. Metabolites and enzymes that were significantly enriched in the PTM group were marked red, those significantly depleted in the PTM group were marked blue, and those with no significant changes were marked black.
4. Functional annotation and visualization: Pathways were categorized by biological relevance using KEGG Mapper. After removing irrelevant metabolites and metabolic enzymes, integrated visualizations of metabolic flux and enzyme expression patterns were generated and presented in figure 7 d-g.

We have included the above information in the Methods part, line 935-945.

Thank you for all the questions as they have improved the quality of our work and the manuscript.

Reviewer #2 (Remarks to the Author)

In this study entitled "snFLARE-seq and mrFRIGID for the transcriptomic and metabolomic landscape of prostate cancer with different anatomical origins" He et al. investigate the molecular differences between prostate tumors sectioned from the peripheral zone (PZ) and the transition zone (TZ). Their research question is of interest to the field and merits attention however the manuscript could be better focused on the important questions and at the moment it is very dense and not so easy to follow. The number of prostatectomy samples included in the study – 270 treatment-naïve and 218 neoadjuvant ADT-treated samples – is impressive and they would be enough to address the question at hand. However, only 14 samples are analyzed for snRNA-seq and metabolites profiling and this limits the impact of the study. The authors report also two novel laboratory protocols, snFLARE-seq and mrFRIGID, that have been used in analyzing these tumor samples.

These ambitious efforts to prove the viability of these two methods, in addition to addressing the biological question of PZ vs TZ characteristics, results in a compromise that provides unfortunately somewhat limited evidence for proving either of the efforts worthy.

Below are our major comments:

(1) One of the findings presented in the manuscript, as stated on lines 139-140, is that "cancers spanning PZ and TZ are more aggressive to resist clinical treatment." Considering that survival analysis with biochemical recurrence as endpoint is the only analysis performed on the complete set of 270 + 218 samples, the presentation of these numbers at the start of results misrepresents the amount of samples used for molecular analyses and should be clarified throughout the manuscript, and to our opinion do not provide sufficient evidence for the conclusions.

Reply:

Thank you for this question. We have revised the manuscript and figures to indicate the numbers for clinical analyses and molecular analyses more clearly (line 151 and 156). In this manuscript, we have developed two new methods and demonstrated the feasibility of an integrative analysis in cancer patients. We are currently undergoing a more comprehensive research with more clinical specimens for molecular analyses, which, due to current funding limitations, might require extended timelines. So, at this stage, this pioneer manuscript is sent out to report the establishment and potential clinical applications of snFLARE-seq and mrFRIGID.

Biochemical recurrence (BCR) is a well-established clinical endpoint in numerous

clinical studies on prostate cancer. Considering the limitations of our single-center retrospective cohort, we would like to amend our final statement, 'The results from our single-center retrospective cohort indicate that cancers spanning PZ and TZ might be more aggressive to resist neoadjuvant ADT', as showing in line 146-148.

We have some remarks on the comparisons that have been used to arrive to the conclusion mentioned above:

①- *Presence of 3+3 GS tumors, notoriously indolent is low in the PZ+TZ, while the same group is enriched with T4 stage PCa.*

Reply:

Thank you for this question. We randomly excluded some GS 3+3 patients from the PZ group and the TZ group, and T4-stage patients from the PZ+TZ group to better match the baselines of the three groups in both the radical and neoadjuvant cohorts. After re-performing the survival analysis, we reached a similar conclusion (**Revised table 1-2; Revised figure 2**). We have included these results as the new **Figure 1** and **Extended data table 1-2**.

Revised table 1: Baseline characteristics of radical cohort with revised Gleason Score.

Variables	1 N = 142	2 N = 59	3 N = 34	p-value ¹
Age, Mean (SD)	69.01 (6.36)	68.51 (7.64)	68.85 (5.42)	0.9
Prostate Volume, Mean (SD)	40.10 (24.85)	38.17 (18.02)	34.23 (15.10)	0.5
PSA, Mean (SD)	10.89 (7.81)	13.47 (11.43)	13.88 (8.43)	0.014
Gleason Score, n (%)				0.4
3+3	23 (16%)	14 (24%)	5 (15%)	
3+4	47 (33%)	26 (44%)	14 (41%)	
4+3	47 (33%)	11 (19%)	10 (29%)	
≥4+4	25 (18%)	8 (14%)	5 (15%)	
T stage, n (%)				<0.05
T1	15 (11%)	5 (8.5%)	1 (2.9%)	
T2a	73 (51%)	17 (29%)	11 (32%)	
T2b	7 (4.9%)	4 (6.8%)	3 (8.8%)	
T2c	14 (9.9%)	5 (8.5%)	8 (24%)	
T3	33 (23%)	28 (47%)	11 (32%)	

¹Kruskal-Wallis rank sum test; Pearson's Chi-squared test

Revised table 2: Baseline characteristics of neoadjuvant cohort with revised Gleason Score.

Variables	1 N = 87	2 N = 31	3 N = 80	p-value ¹
Age, Mean (SD)	68.92 (6.94)	71.23 (4.20)	68.61 (6.09)	0.2
Prostate Volume, Mean (SD)	41.18 (16.75)	45.52 (21.15)	42.36 (20.18)	0.6
PSA, Mean (SD)	73.51 (119.11)	48.90 (35.36)	70.39 (65.91)	0.3
Gleason Score, n (%)				0.9
≤3+4	3 (3.5%)	3 (9.7%)	5 (6.3%)	
4+3	16 (19%)	7 (23%)	15 (19%)	
4+4	43 (50%)	14 (45%)	41 (51%)	
≥4+5	24 (28%)	7 (23%)	19 (24%)	
T stage, n (%)				0.12
≤T3a	29 (33%)	18 (58%)	29 (36%)	
T3b	45 (52%)	8 (26%)	39 (49%)	
T4	13 (15%)	5 (16%)	12 (15%)	

¹Kruskal-Wallis rank sum test; Fisher's exact test

②- While perhaps not explicitly stated before, it is widely recognised that larger tumors, i.e. tumor spanning multiple anatomical regions have more cancer cells and are consequently "more aggressive" and predisposed to be more treatment-resistant than smaller tumors.

Reply:

Thank you for this question. That might be an explanation to the aggressiveness of tumors spanning PZ and TZ. Therefore, we re-examined the postoperative pathological reports of patients with localized prostate cancer cohort (**Figure 1a**) and compared the maximum diameter of the index tumors. We found that tumors spanning PZ and TZ are the largest, while those zone originating from PZ are the smallest (PZ : TZ : PZ+TZ = 2 cm : 2.4 cm : 3 cm, P < 0.05). Univariate Cox regression analysis revealed that maximum diameter of the index tumor was negatively correlated with biochemical recurrence

(BCR) outcomes in patients (HR 1.624, 95% CI 1.135 - 2.325, P < 0.05), which is consistent with general understanding. However, a multivariate Cox regression analysis incorporating factors such as age, prostate volume (PV), Gleason score (GS), T stage, and zone, found no significant association between maximum diameter of the index tumor and BCR survival (P = 0.236) (**Revised table 3**).

Revised table 3: Multivariate Cox regression analysis.

Variable	Hazard Ratio (95% CI)	P-value
Maximum diameter of index tumor	1.29 (0.85-1.95)	0.236
Zone		
PZ	Reference	
TZ	0.21 (0.06-0.74)	0.015
PZ+TZ	1.81 (0.76-4.28)	0.180
Age	0.96 (0.91-1.02)	0.151
Prostate Volume	0.98 (0.96-1.00)	0.101
Prostate Specific Antigen	1.01 (0.96-1.05)	0.816
T stage		
T1	Reference	
T2	0.38 (0.08-1.78)	0.218
T3	1.57 (0.34-7.25)	0.567
Gleason Score		
3+3	Reference	
3+4	2.10 (0.58-7.56)	0.257
4+3	1.76 (0.46-6.74)	0.406
≥4+4	2.70 (0.67-10.91)	0.163

It has been reported that patients with TZ prostate cancer are characterized by higher

baseline prostate-specific antigen (PSA) levels and larger cancer volume, but showed more favorable clinical outcomes than those with PZ cancer (8). Therefore, it seems that tumor size, while being an important clinic indicator, might not be that crucial in determining tumor aggressiveness.

③- *Genetics status of the tumors, which can also determine a more aggressive tumor phenotype is not taken into account: e.g. PTEN loss, P53 and ERG-fusion status, are just few of the obvious alterations associated with aggressive phenotypes and these should be accounted for in the analysis.*

Reply:

Thank you for this question. We fully agree that the genetic status of tumors is intricately linked to phenotypes like invasiveness and drug resistance. Unfortunately, at our center, genetic testing is currently not included in the routine diagnostic protocol for patients undergoing radical prostatectomy, with or without neoadjuvant therapy.

In this study, we delved into transcriptomics at the single-cell level. Traditional bulk genomic sequencing fails to account for the transcriptomic heterogeneity observed at the single-cell level. Given that different cells may harbor distinct genetic mutations, the development of single-cell genome sequencing technologies is imperative to elucidate the correlation of genetic mutations with transcriptomic alterations. This unmet challenge attracts the interest of numerous talented scientists. We sincerely wish to conduct such an analysis in the near future. Notably, compared with Western patients, Chinese prostate cancer patients exhibit a lower frequency of PTEN deletion and ERG-fusion (9, 10).

④- *Biochemical recurrence is a surrogate endpoint and in the analysis the tumor are characterised for their index lesion only. As prostate cancer can also be multifocal, lymph nodes molecular characterisation might offer a more accurate endpoint to delineate "zone-related tumor heterogeneity and its correlation with disease aggressiveness"*

Reply:

Thank you for this question. Biochemical recurrence (BCR) serves as a widely adopted

surrogate endpoint for metastasis-free survival (MFS) in numerous prostate cancer clinical studies. MFS and overall survival (OS) require long-term follow-up period. Observing lymph node metastasis demands an extended follow-up period and obtaining the molecular characteristics of lymph nodes poses significant challenges for retrospective analysis. In light of the current medical situation in our country and the high mobility of patients, the feasibility of selecting other indicators for analysis is challenging. Therefore, BCR was selected as the study endpoint.

We try our best to reevaluate the lymph node recurrence status in our two clinical cohorts. As anticipated, no lymph node metastasis was detected in the radical cohort. In the neoadjuvant cohort, 16 cases of lymph node metastasis were noted: 5 in the PZ group, 3 in the TZ group, and 8 in the PZ+TZ group. We also conducted a survival analysis with lymph node metastasis as the endpoint. However, due to the limited metastasis events, no statistically significant differences were observed (**Revised figure 3**).

Index lesion is widely used for multifocal prostate cancer evaluation, including TNM staging, Gleason scoring, risk stratification, and treatment selection. Index foci accounts for 60-80% of the total prostate cancer burden and serves as the primary driver of disease progression. Evaluation based on index lesion would reduce pathological assessment complexity. Multiple clinical trials (e.g., SUMPRO, RADAR) confirm that pathological analysis centered on index foci demonstrates higher

consistency with clinical outcomes than multifocal assessment. The 2019 European Association of Urology (EAU) Guidelines for Prostate Cancer explicitly recommend prioritizing the description of index foci features as the primary basis for risk stratification.

(2) Effective utilisation of FFPE preserved biological material is a major hurdle in not only prostate cancer research, but in every context where FFPE-preserved material is used. Novel, improved methods for extracting RNA and metabolites would have be in high demand. With regards to RNA, the authors reference existing alternatives snPATHO-seq and snRandom-seq in their introduction, but only provide a very brief benchmark of their snFLARE-seq method against snRandom-seq. Rather, the authors use a variety of tissues to conclude that "All tissue type samples achieved a median genes per cell count of above 900, indicating the robustness of snFLARE-seq across various malignancies" (205-207). This result provides no significant improvement to snPATHO-seq (PMID:39414943), although snFLARE-seq is still worthy of further development for investigation of archival material. As per the evidence shown, more benchamarks are needed, to demonstrate which method could be used to obtain better results on archival material. We would therefore suggest reworking the title of the study, focusing on the biological observations derived from the investigation of the different anatomical compartments of the prostate, or on the description and benchmark of the novel methods snFLARE-seq and mrFRIGID.

Reply:

Thank you for this question. As noted by the reviewers, each technique has its unique characteristics and applicable scenarios. The selection of a specific technique not only depends on the scientific question at hand but is also influenced by factors such as technical accessibility and cost. Therefore, this paper does not aim to emphasize that our method outperforms others in all aspects, a claim that might encounter unnecessary resistance for the publication. Instead, it highlights the development of an accessible, cost-effective, distinctive, and reliable technique for single-cell transcriptomic analysis of paraffin-embedded samples. Additionally, it explores the potential of integrating transcriptomic data with metabolomics to leverage FFPE samples for scientific discovery.

As suggested by the reviewer, we have incorporated additional benchmarking analyses. Here we conduct snRNA-sequencing analysis using lung cancer FFPE samples. The snFLARE-seq protocol successfully captured 7019 cells, with a median of 1278

genes per nucleus (sequencing saturation: 20.55%). Compared with probe-based scFFPE (sample ID: Lung_Cancer_Manual_BC), snFLARE-seq exhibited superior gene

detection capacity, as evidenced by the increased gene counts per cell (**Revised figure 4A**). In terms of cellular composition, epithelial cells dominated the clusters identified by scFFPE-seq, whereas snFLARE-seq revealed a broader spectrum of immune cells and fibroblasts (**Revised figure 4B**). These discrepancies might be attributed to inherent heterogeneity of these samples, but at least these results demonstrate that snFLARE-seq achieves higher transcriptomic resolution per cell while maintaining balanced detection across diverse cell types.

To compare with snPATHO-seq protocol, we conducted snFLARE-seq with formaldehyde-fixed PBMC samples. Both methods yielded comparable cellular distributions (**Revised figure 4C**). However, snFLARE-seq employs random primers and oligo d(T) for cDNA library preparation, enabling comprehensive exon coverage of transcribed genes (**Revised figure 4D**). This design facilitates downstream applications such as alternative splicing analysis and single-nucleotide variant (SNV) detection (**Revised figure 4E**). We have incorporated part of these results in Extended data figure 4.

(3) The authors report snRNA-seq results from 101,729 cells extracted from 14 samples and 9 patients (5 treatment-naïve, 4 neoadjuvant-treated). The results recapitulate previously described epithelial cell subtypes, including tumor, luminal, basal, club, hillock, and neuroendocrine cells. These data are novel, considering that previous prostate scRNA-seq datasets have been produced exclusively from the fresh or fresh frozen tissue. However, the consequent analysis is hampered by the small number of patients and samples, especially considering the wide range of conditions covered (TZ, PZ, treatment-naïve, neoadjuvant ADT-treated). The authors present a finding that club cells are enriched post-treatment. Further analysis regarding the epithelial cells are limited to presenting enriched gene ontology pathways with little to limited interpretation of the underlying biology.

Reply:

Thank you for this question.

This study aims to develop methods for FFPE samples to expand their utility beyond clinical validation and integrate them into scientific discovery cohorts. With these methods, we seek to address scientific questions that remain unresolved due to the limitations of current research models in replicating real-world clinical patient

scenarios.

Currently there is lack of suitable biology model to mimic human prostate cancer of different zone origins, and we utilized our methods to perform a pilot investigation using limited clinical samples. Our findings reveal that integrating FFPE-based single-cell sequencing with metabolomic profiling can uncover zonal-specific heterogeneity in prostate cancer, along with associated metabolic pathways and molecular targets. We also discovered the remodeling effects of hormone therapy on prostate cancer cells and their microenvironment.

We agree with the reviewer and validate our findings with external datasets. As shown in Revised figure 5, club cells were also enriched in prostate cancer regions but not in the healthy patients or noncancerous adjacent areas (**Revised figure 5A**)(11). In another study comparing the early onset prostate cancer (EOPC, which is recognized more aggressive) and late onset prostate cancer (LOPC), club cells were also enriched in EOPC but not LOPC (**Revised figure 5B**)(12). We have included these results in the revised Extended data Fig 6 c and d.

As stated above, currently there is lack of suitable biology model to mimic human prostate cancer of different zone origins, or club cells, making it difficult to perform biological mechanistic investigation. However, we still try our best to investigate club cells using data-driven approaches. In figure 3 h-g, we utilized Dynamic network

biomarkers (DNB) analysis, a unique algorithm developed by our team to find out pinpointing fate-decision points (3, 4, 13-18), to find out that PI3K-AKT signaling and choline related metabolism are crucial for disease progression and aggressive evolution.

Because models that mimic peripheral zone or transition zone prostate cancer are currently unavailable, more biological validation could be not performed. Please refer to literatures from Nature Genetics, 2025 and Clin Transl Med, 2022, researchers also mainly performed data-driven analysis (19, 20). If there are models (cell lines, organoids, or mouse models) for mechanistic investigation of zone specific heterogeneities, we would use fresh samples for scRNA-seq and metabolomics investigation, but not these complicate approaches. We choose a strategy to separate the discovery phase results from mechanistic investigation: using FFPE samples to find out potential critical pathways and genes associated with disease aggressiveness; and then investigate the related **conserved** mechanisms in other systems including prostate cancer cell lines and organoids (mainly from CRPC patients) , to elucidate the association of the identified pathways and genes with disease aggressiveness. These two parts should logically be presented separately rather than merged in a single paper. At this stage, we present our initial results as a solid starting point for further investigation.

Moving forward, our team will expand the cohort to identify clinically relevant disease heterogeneity and progression mechanisms. These efforts will be complemented by in-depth mechanistic studies and drug discovery initiatives. The systematic characterization of these mechanisms represents one of our long-term research goals.

(4) The enrichment of exhausted T cells, regulatory T cells, and M2-polarized macrophages post-treatment is an interesting and novel observation. Unfortunately this analysis is hampered by the low number of samples from which the single data has been generated from, making it difficult to generalise this result. In this case, datamining currently available dataset may corroborate the data, however, would still be limited in consideration of the conditions that should be covered (TZ, PZ, treatment-naïve, neoadjuvant ADT-treated) to provide some additional evidence.

Reply:

Thank you for this question. We now validate our findings using the same external databases as referred above (11, 12). As shown in Revised figure 6, exhausted T cells and Treg cells were enriched in prostate cancer regions but not in the healthy patients

or noncancerous adjacent areas (**Revised figure 6A**)(11). In another study comparing the early onset prostate cancer (EOPC, which is recognized more aggressive) and late onset prostate cancer (LOPC), exhausted T cells and Treg cells were also enriched in EOPC but not LOPC (**Revised figure 6B**)(12). Similar results were observed for M2 macrophages in these two databases (**Revised figure 6 C and D**). These data are consistent with our findings and have been incorporated into the revised manuscript as Extended data Fig.13 and Extended data Fig.15 e and f.

Minor comments

(1) Placement of figures and figure legends significantly worsens the readability of the manuscript.

Reply:

Thank you for this suggestion. We have revised it accordingly.

(2) mrFRIGID method is another important novel, potentially improved method for extracting valuable information from archival material. The reviewers did not accurately review it since we feel we did not have enough expertise. We however feel that the name of this method could be changed to avoid contextual misinterpretations.

Reply:

Thank you for this suggestion. The mrFRIGID method, building upon the F-method, incorporates enhancements in the number of extraction steps and the separation of hydrophilic-lipophilic layers. To avoid confusion with the mrFRIGID method, we have renamed F-method, H-method, and X-method as **F-extraction**, **H-extraction**, and **X-extraction**, respectively.

Thank you for all the questions as they have improved the quality of our work and the manuscript.

Reviewer #5, expertise in single cell sequencing method development (Remarks to the Author):

This study presents a new multi-omics approach to investigate the heterogeneity of prostate cancer originating from different anatomical zones and its evolution under hormone therapy. The authors developed two novel methodologies, snFLARE-seq for single-nucleus transcriptomics and mrFRIGID for metabolomics, tailored for FFPE samples, addressing longstanding technical challenges in analyzing archived clinical specimens. By integrating these techniques with clinical data, the work reveals distinct transcriptomic and metabolic profiles between zone-specific tumors, identifies therapy-induced remodeling of epithelial cells and immune microenvironments, and links key pathways such as PI3K-AKT and choline metabolism to disease aggressiveness. The cohort analyses provide clinically relevant insights, such as poorer outcomes for tumors spanning multiple zones and the role of androgen receptor signaling in treatment resistance. However, limitations in sample diversity, mechanistic depth, and technical transparency hinder broader applicability.

1. The study relies on a small cohort with only nine patients for omics analysis and 270 for clinical data, raising concerns about statistical power and generalizability. All patients appear to be East Asian, which introduces potential bias given known ethnic disparities in prostate cancer biology. For example, African ancestry populations exhibit higher mutational burdens and distinct molecular subtypes not represented here. Expanding cohorts to include diverse ethnicities and larger sample sizes would enhance robustness. Additionally, the exclusion of patients without postoperative PSA decline may omit atypical cases, limiting exploration of outlier biology. Independent validation in multiethnic cohorts is critical to confirm the universality of zone-specific signatures.

Reply:

Thank you for this question. It has been well documented that prostate cancer originating from the peripheral zone is more aggressive across different cohorts in Caucasian patients and African ancestry population. However, epidemiological characteristics in East Asian populations have remained unexplored. Our investigation now addresses this critical knowledge gap.

We have also reached out to our collaborators in the US for a multi-center and multi-ethnic analysis to meet the reviewer's demand. Regrettably, potential collaborators have expressed reservations about engaging in translational research collaborations with Chinese institutions under current geopolitical circumstances. Also, both our international colleagues and our team acknowledge the scientific merit of such a multi-center and multi-ethnic analysis. This huge project might be initiated at a

suitable time point for a manuscript for NEJM or Lancet Oncology.

For patients without postoperative PSA decline, first, the number of these patients is limited. Second, they typically obtain positive surgical margins and require different and individual clinical treatments for disease management. Consequently, it is not suitable to incorporate them into our cohort.

2. While the study identifies PI3K-AKT and choline metabolism as critical pathways, functional validation is limited to computational predictions and basic cell proliferation assays. The proposed role of hyperforin in targeting IL-17 pathways lacks in vivo evidence or detailed mechanistic studies. Orthogonal approaches such as CRISPR-based gene editing, organoid models, or murine xenografts are needed to confirm causality. Similarly, metabolic dependencies like CHKB-mediated phosphatidylcholine synthesis require isotope tracing or pharmacological inhibition to establish their necessity in tumor progression. Without such validation, the therapeutic potential of these targets remains speculative and risks misdirecting future research.

Reply:

Thank you for this question.

The function of PI3K-AKT, IL-17, and choline metabolism in prostate cancer have been investigated previously in different prostate cancer cell lines and mouse models. Genetically activated PIK3CA in mouse drives the development of prostate cancer (21), and inhibition of PI3K suppresses prostate cancer progression in both Pten-deficient murine model and in human prostate cancer xenografts (22). For IL-17 pathway, the knockout of IL17 receptor, Il17rc, inhibits the development of prostate cancer in mouse model (23). Mechanistically, IL17 pathway may regulate PD-L1 expression in prostate cancer or create an immune-tolerant tumor microenvironment to facilitate the progression of prostate cancer at different disease stages (24-27). For choline metabolism, the essential metabolic enzyme CHKA has been identified as a clinical relevant androgen receptor target in prostate cancer (28) and potential target for prostate cancer therapy (29). With these abundant, exceptional, and well-substantiated data showing the importance of PI3K, IL17, and choline metabolism in prostate cancer, the suggested experiments have been previously performed by other investigators and it is not necessary to rely on our redundant functional validation to

confirm the significance of these pathways. Our results highlight the clinical scenarios associated with these key factors and mechanisms. To meet the reviewer's request to the best of our ability, we have incorporated these literatures in the revised manuscript (line 289-291; line 311-314). Also, we have performed experiments to show that choline promotes cell growth in C4-2 and Du145 cell lines, as shown in Extended data Fig. 21.

Our data here provide clinical evidence to show the importance of these pathways and the application of related inhibitors in specific scenarios, zone-specific prostate cancer receiving adjuvant therapy. Models replicating prostate cancer subtypes specific to the peripheral zone or transition zone remain unavailable, precluding direct biological validation of zone heterogeneity mechanisms. Please refer to literatures from Nature Genetics, 2025 and Clin Transl Med, 2022, researchers also mainly performed data-driven analysis (19, 20). Should validated models emerge (eg., zone-specific cell lines, organoids, or mouse models), we would prioritize fresh samples for scRNA-seq and metabolomics profiling, rather than relying on FFPE samples and these complicated approaches.

In our manuscript, we briefly investigated inhibitors of the IL-17 pathway in LNCaP, C4-2, and VCaP Cell lines in Extended data figure 9. Actually, these cell lines are from CRPC patients and it is scientifically inconsistent in explanations of primary zonal pathogenesis. That is why we place these results in Extended data but not in the main figures. Similarly, using current models to validate PI3K and choline metabolism function in zone-specific prostate cancer is also scientifically inconsistent. It is unnecessary to present excessive flawed data only to show our unprofessionalism to colleagues in the prostate cancer field.

Moving forward, we would strategically decouple discovery-phase analyses from mechanistic validation for scientific questions short of ideal biology models. We first leveraged FFPE archives to identify candidate pathways (e.g., PI3K/AKT signaling) and metabolic signatures (e.g., choline metabolism) correlated with disease aggressiveness in the discovery phase. Then, we would conduct mechanistic validation

through established cell lines and organoids, despite their inherent limitations of originating from CRPC metastatic sites rather than primary zone lesions, to verify the likely **conserve** functional causality of pathways and metabolic signatures to disease aggressiveness. The mechanistic validation research would find novel target and drugs for prostate cancer management, while the discovery-phase analyses would indicate the clinical scenarios for these drugs/treatments. At this stage, we present our initial results as a solid starting point for further investigation.

3. The analysis of immune cell subtypes lacks spatial context, a critical gap given the importance of localized cell interactions in therapy resistance. Single-cell RNA-seq data alone cannot resolve spatial relationships between epithelial cells, fibroblasts, and immune populations. Techniques like spatial transcriptomics or multiplex immunofluorescence would clarify whether enriched Tregs or M2 macrophages localize to specific tumor niches. Furthermore, the absence of functional immune assays such as T-cell cytotoxicity or macrophage phagocytosis limits the ability to corroborate transcriptomic signatures of exhaustion or suppression.

Reply:

Thank you for this suggestion. We have performed multiplex immunofluorescence with our FFPE samples. For TZ samples, more CD8⁺ T cells were identified, but mainly in the stromal area rather than the tumor area. Infiltration of immune cells was more frequently found in PZ samples. For PTM samples, less CD8⁺ T cells and more M2 macrophages were identified; PD1⁺ cells increased. Interestingly, there is rich infiltration of immune cells in PTM samples, but most of these immune cells are associated with immunosuppression. We have included these data as **Figure 4h** and **Extended Data table 3**.

Immunologists have dedicated substantial efforts to identifying markers that characterize the functions of immune cells, to avoid repeatable functional validation. We are interested in investigating novel immune cell subpopulations in prostate cancer patients. However, it is challenging to collect immune cells from the tumor microenvironment of prostate cancer patients for functional immune assays. Different from other tumors, prostate cancer can hardly be distinguished from adjacent noncancerous area without HE staining. Ideally, we should collect fresh specimens

from prostate cancer patients and hold it to perform HE staining with two separated sides (bottom side and top side) to locate potential tumor areas for immune cell enrichment (Revised figure 7). It requires close coordination of our clinicians, pathologist, and scientific researchers, after the approval of a new ethnic protocol. Our team for this manuscript locate in three different cities, making it more challenging to execute this task currently.

4. The study does not fully address technical biases inherent to snFLARE-seq, despite its dual use of random and oligo-dT primers. While the method demonstrates improved gene detection in FFPE samples, its performance on non-FFPE samples such as cell lines or fresh tissues remains unclear. For instance, how does snFLARE-seq compare to mainstream 3' scRNA-seq platforms like 10X Genomics in detecting exon versus intronic reads or integrating with existing datasets? The proportion of intronic reads, which may reflect pre-mRNA capture, is not quantified, potentially complicating comparisons with polyA-selected datasets. Additionally, the method's bias toward longer RNA fragments or genes due to FFPE-induced degradation is not discussed. Systematic benchmarking against gold-standard fresh-tissue protocols and detailed reporting of read distribution across gene bodies would help users assess compatibility with their research goals.

Reply:

Thank you for this suggestion.

Our snFLARE-seq is developed to leverage the rich FFPE sample resources for generating or discovering novel scientific hypotheses, extending beyond their conventional role in clinical validation. Thus, snFLARE-seq is specifically optimized for

FFPE samples or formalin-fixed samples, but could not be used directly for fresh samples. Alternative transcriptomic platforms are recommended for fresh cell lines or tissues. We also compared the performance of snFLARE-seq with FFPE samples and snRNA-seq with fresh tissues. As shown in **Extended Data Fig. 3**, two fresh mouse kidney samples and two FFPE mouse kidney samples were used for analysis. A median of 817 and 818 genes per nucleus were detected in fresh kidney samples, with sequencing saturation at 41.62% and 43.10%, respectively. snFLARE-seq yielded a median of 786 and 578 genes per nucleus from FFPE samples, with sequencing saturation at 38.34% and 38.25%, respectively (**Extended Data Fig. 3b**). The unique molecular indices (UMIs) counts were also found to be comparably between the two methods (**Extended Data Fig. 3b**). A total of 16,119 cells were clustered into 15 distinct cell types (**Extended Data Fig. 3 c and d**), and the cell types identified in FFPE kidney samples were largely consistent with those determined in fresh kidney samples (**Extended Data Fig. 3 e and f**). Additionally, the gene expression profiles obtained from two different samples were found to be relatively consistent (**Extended Data Fig. 3g**).

We also compared our snFLARE-seq –derived PBMC data (PBMC-snFLARE-seq) to public available PBMC results using scRNA-seq (PBMC-scRNA-seq; <https://www.10xgenomics.com/datasets/1-k-pbm-cs-from-a-healthy-donor-v-3-chemistry-3-standard-3-0-0>) and snRNA-seq (PBMC-snRNA-seq; <https://www.10xgenomics.com/datasets/pbmc-from-a-healthy-donor-no-cell-sorting-3-k-1-standard-2-0-0>) from 10 X Genomics. Notably, both 10 X Genomics datasets were generated using oligo dT primers. The median numbers of genes detected per cell were 1250 (PBMC-snFLARE-seq), 1949 (PBMC-scRNA-seq), and 1494 (PBMC-snRNA-seq), consistent with the known lower RNA content in nuclear compared to whole-cell analyses (**Revised figure 8A**).

Comparative analysis revealed markedly elevated intronic read proportions in snFLARE-seq (85.2%) versus PBMC-scRNA-seq (35.5%) and PBMC-snRNA-seq (49.0%) (**Revised figure 8B**). This discrepancy could be attributed to the application of random primers in snFLARE-seq. The intron information is helpful for analyses on alternative

splicing and neo-antigens. Consistently, snRandom-seq, another random primers-based method, demonstrates comparable intronic read proportions (75%) in 293T and 3T3 cells (30).

Notwithstanding these technical differences, cross-platform correlation analysis between PBMC-snFLARE-seq and PBMC-snRNA-seq demonstrated robust linear agreement (Pearson's $r = 0.82$, $p < 2.2 \times 10^{-16}$), indicating substantial technical reproducibility in gene detection across platforms (**Revised figure 8C**). We further conducted integrative analysis of these three independent single-cell sequencing datasets and systematically evaluated cross-platform comparability. Through joint embedding in a unified low-dimensional space, phenotypically matched cell populations—including CD16⁺/CD14⁺ monocytes, hematopoietic stem cells (HSC), dendritic cells (DC), T/B lymphocytes, and plasma cells- exhibited highly concordant clustering patterns across distinct experimental systems in the UMAP projection (**Revised figure 8D**). These results demonstrate that standardized integration pipelines effectively minimize technical variance and maintain biological fidelity, achieving cross-dataset functional annotation consistency and thereby proving the

feasibility of robust multi-technology data harmonization.

Contrary to the prevailing consensus that RNA extracted from FFPE samples is intrinsically fragmented, we discovered that pre-decrosslinking treatment of nuclei prior to RNA extraction yields substantially longer RNA fragments. This indicates that a portion of intact RNA within FFPE cells becomes fragmented during the extraction process. Consequently, we hypothesize that formaldehyde-induced protein-RNA crosslinks generate steric hindrance during reverse transcription, leading to truncated cDNA products in certain methods, such as snRANDOM-seq. Guided by this mechanistic insight, we developed the snFLARE-seq workflow with integrated pre-decrosslinking, which enables: 1. Detection of extended RNA transcripts (**Extended Data Fig. 4 c-d**); 2. Increase in median gene detected per nucleus (**Extended Data Fig. 2a ; Extended Data Fig. 4e**); 3. Compatibility with diverse library preparation methods (**Extended Data Fig. 4h**). Notably, these improvements do not support the purported "long RNA fragment bias" in snFLARE-seq. Rather, the pre-decrosslinking step itself unlocks the access to longer nuclear transcripts. The read distribution across gene bodies has been shown in **Extended Data Fig. 4f**.

Through systematic quality control of FFPE-derived RNA across diverse tissue sources, we identified DV200 and total RNA quantity as two critical parameters correlating with snFLARE-seq success rates and gene detection. Thus, we established standardized quality thresholds : total RNA from 20,000 nuclei \geq 25 ng, and predominant RNA fragments is \geq 200 nucleotides (**Extended Data Fig. 4i**).

Thank you for all the questions as they have improved the quality of our work and the manuscript.

Reference:

1. Zheng L, *et al.* (2021) Pan-cancer single-cell landscape of tumor-infiltrating T cells. *Science* 374(6574):abe6474.
2. Zhang X, *et al.* (2024) Multi-omics with dynamic network biomarker algorithm prefigures organ-specific metastasis of lung adenocarcinoma. *Nature communications* 15(1):9855.
3. Yang B, *et al.* (2018) Dynamic network biomarker indicates pulmonary metastasis at the tipping point of hepatocellular carcinoma. *Nature communications* 9(1):678.
4. Gao R, *et al.* (2024) mNFE: microbiome network flow entropy for detecting pre-disease states of type 1 diabetes. *Gut microbes* 16(1):2327349.
5. Chen C, *et al.* (2024) Explore key genes of Crohn's disease based on glycerophospholipid metabolism: A comprehensive analysis Utilizing Mendelian Randomization, Multi-Omics integration, Machine Learning, and SHAP methodology. *International immunopharmacology* 141:112905.
6. Sganzerla Martinez G, *et al.* (2024) Identification of Marker Genes in Infectious Diseases from ScRNA-seq Data Using Interpretable Machine Learning. *International journal of molecular sciences* 25(11).
7. Goel U, Usmani S, & Kumar S (2022) Current approaches to management of newly diagnosed multiple myeloma. *American journal of hematology* 97 Suppl 1:S3-S25.
8. Sato S, Kimura T, Onuma H, Egawa S, & Takahashi H (2021) Transition zone prostate cancer is associated with better clinical outcomes than peripheral zone cancer. *BJUI compass* 2(3):169-177.
9. Li J, *et al.* (2020) A genomic and epigenomic atlas of prostate cancer in Asian populations. *Nature* 580(7801):93-99.
10. Dong B, *et al.* (2024) Integrative proteogenomic profiling of high-risk prostate cancer samples from Chinese patients indicates metabolic vulnerabilities and diagnostic biomarkers. *Nature cancer* 5(9):1427-1447.
11. Hirz T, *et al.* (2023) Dissecting the immune suppressive human prostate tumor microenvironment via integrated single-cell and spatial transcriptomic analyses. *Nature communications* 14(1):663.
12. Cheng Y, *et al.* (2025) Single-cell and spatial RNA sequencing identify divergent microenvironments and progression signatures in early- versus late-onset prostate cancer. *Nature aging*.
13. Aihara K, Liu R, Koizumi K, Liu X, & Chen L (2022) Dynamical network biomarkers: Theory and applications. *Gene* 808:145997.
14. Chen L, Liu R, Liu ZP, Li M, & Aihara K (2012) Detecting early-warning signals for sudden deterioration of complex diseases by dynamical network biomarkers. *Scientific reports* 2:342.
15. Liu X, *et al.* (2019) Detection for disease tipping points by landscape dynamic network biomarkers. *National science review* 6(4):775-785.
16. Li L, *et al.* (2023) Dynamic network biomarker factors orchestrate cell-fate determination at tipping points during hESC differentiation. *Innovation*

- 4(1):100364.
17. Fang Z, *et al.* (2023) Oxidative stress-triggered Wnt signaling perturbation characterizes the tipping point of lung adeno-to-squamous transdifferentiation. *Signal Transduct Target Ther* 8(1):16.
 18. Tong Y, *et al.* (2023) Earthquake alerting based on spatial geodetic data by spatiotemporal information transformation learning. *Proceedings of the National Academy of Sciences of the United States of America* 120(37):e2302275120.
 19. Hu J, *et al.* (2025) Spatially resolved transcriptomic analysis of the adult human prostate. *Nature genetics* 57(4):922-933.
 20. Yan Q, *et al.* (2022) Single-cell RNA-sequencing technology demonstrates the heterogeneity between aged prostate peripheral and transitional zone. *Clinical and translational medicine* 12(10):e1084.
 21. Pearson HB, *et al.* (2018) Identification of Pik3ca Mutation as a Genetic Driver of Prostate Cancer That Cooperates with Pten Loss to Accelerate Progression and Castration-Resistant Growth. *Cancer discovery* 8(6):764-779.
 22. Carver BS, *et al.* (2011) Reciprocal feedback regulation of PI3K and androgen receptor signaling in PTEN-deficient prostate cancer. *Cancer cell* 19(5):575-586.
 23. Zhang Q, *et al.* (2012) Interleukin-17 promotes formation and growth of prostate adenocarcinoma in mouse models. *Cancer research* 72(10):2589-2599.
 24. Lin G, *et al.* (2024) IL-17RA/CTSK axis mediates H. pylori-induced castration-resistant prostate cancer growth. *Oncogene* 43(49):3598-3616.
 25. Wang X, *et al.* (2017) Inflammatory cytokines IL-17 and TNF-alpha up-regulate PD-L1 expression in human prostate and colon cancer cells. *Immunology letters* 184:7-14.
 26. Zhang Q, *et al.* (2017) Targeting Th17-IL-17 Pathway in Prevention of Micro-Invasive Prostate Cancer in a Mouse Model. *The Prostate* 77(8):888-899.
 27. Zhang Q, *et al.* (2014) Interleukin-17 promotes development of castration-resistant prostate cancer potentially through creating an immunotolerant and pro-angiogenic tumor microenvironment. *The Prostate* 74(8):869-879.
 28. Asim M, *et al.* (2016) Choline Kinase Alpha as an Androgen Receptor Chaperone and Prostate Cancer Therapeutic Target. *Journal of the National Cancer Institute* 108(5).
 29. Wen S, *et al.* (2020) Aberrant activation of super enhancer and choline metabolism drive antiandrogen therapy resistance in prostate cancer. *Oncogene* 39(42):6556-6571.
 30. Xu Z, *et al.* (2023) High-throughput single nucleus total RNA sequencing of formalin-fixed paraffin-embedded tissues by snRandom-seq. *Nature communications* 14(1):2734.